# Synthetic Peptides as a Promising Alternative to Control Viral Infections in Atlantic Salmon

**DOI:** 10.3390/pathogens9080600

**Published:** 2020-07-23

**Authors:** Constanza Cárdenas, Fanny Guzmán, Marisela Carmona, Cristian Muñoz, Luis Nilo, Alvaro Labra, Sergio H. Marshall

**Affiliations:** 1Núcleo Biotecnología Curauma, Pontificia Universidad Católica de Valparaíso, Valparaíso 2373223, Chile; fanny.guzman@pucv.cl; 2Laboratorio de Genética e Inmunología Molecular, Instituto de Biología, Pontificia Universidad Católica de Valparaíso, Valparaíso 2373223, Chile; marisela.carmona@gmail.com (M.C.); cristian.m.bravo@gmail.com (C.M.); 3Facultad de Ingeniería y Ciencias, Universidad Adolfo Ibañez, Santiago 7910000, Chile; luis.nilo@uai.cl; 4Laboratorio de Patogenos Acuícolas, Pontificia Universidad Católica de Valparaíso, Valparaíso 2373223, Chile; alvaro.labra@pucv.cl

**Keywords:** interfering peptides, viral treatment, RNA fish viruses

## Abstract

Viral infections in salmonids represent an ongoing challenge for the aquaculture industry. Two RNA viruses, the infectious pancreatic necrosis virus (IPNV) and the infectious salmon anemia virus (ISAV), have become a latent risk without healing therapies available for either. In this context, antiviral peptides emerge as effective and relatively safe therapeutic molecules. Based on in silico analysis of VP2 protein from IPNV and the RNA-dependent RNA polymerase from ISAV, a set of peptides was designed and were chemically synthesized to block selected key events in their corresponding infectivity processes. The peptides were tested in fish cell lines in vitro, and four were selected for decreasing the viral load: peptide GIM182 for IPNV, and peptides GIM535, GIM538 and GIM539 for ISAV. In vivo tests with the IPNV GIM 182 peptide were carried out using *Salmo salar* fish, showing a significant decrease of viral load, and proving the safety of the peptide for fish. The results indicate that the use of peptides as antiviral agents in disease control might be a viable alternative to explore in aquaculture.

## 1. Introduction

Aquaculture is an important industry in Chile, with at least three species of farmed salmon, Coho salmon (*Oncorhynchus kisutch*), rainbow trout (*Oncorhynchus mykiss*) and Atlantic salmon (*Salmo salar*), cultivated in net pens with the latter being the second most valuable export product of the country, after copper.

Fish farming involves great risks at the sanitary level, with the development of pathogens that seriously affect the sustainability of the industry. Viral infections are among these pathogenic threats, particularly in Atlantic salmon where two viruses, Infectious Pancreatic Necrosis Virus (IPNV) and Infectious Salmon Anemia Virus (ISAV), represent a continuous challenge in production.

IPNV is a member of the *Birnaviridae* family and the etiological agent of infectious pancreatic necrosis, one of the main diseases that affect salmonid fish in general; in Chile, it represents the second cause of infectious mortality in fry [1]. Although it does not currently represent a high risk, survivors of infection become asymptomatic carriers of the virus for life, acting as reservoirs and spreaders of progeny viruses through the water via feces mainly during stress episodes, and as breeders, through their reproductive products [2]. The disease occurs mainly in freshwater, affecting fish in the fry, pre-smolt, and smolt stages of development, when the immunological response of fish is depressed, or in the brood stock fish with the risk of vertical transmission [3,4]. On the other hand, some animals that recover from infection become labile which decreases productivity and health performance.

IPNV is a double-stranded RNA non-enveloped virus, whose genome comprises two segments: segment A with two ORFs, the first one encoding the non-structural protein of 17 kDa VP5, and the second one encoding a polyprotein of 107 kDa, which after processing generates the proteins pVP2, VP3 and VP4 (54, 25 and 28 kDa, respectively); and segment B which encodes the RNA-dependent RNA polymerase, a protein of 97 kDa. Structurally the virion enclosing VP1, genomic RNA, and VP3 are assembled into an icosahedral capsid formed by VP2, and the initial step in the process of virus encapsidation seems to be the generation of VP2 (or pVP2) protein homotrimers, as building blocks [5,6,7]. 

ISAV is the agent of infectious salmon anemia and belongs to the *Orthomyxoviridae* family. ISAV was the cause of major outbreaks in the Atlantic salmon industry in Chile between 2007 and 2010, with significant losses both from a social and from an economic perspective. Currently, the situation has improved thanks to adequate sanitary management; however, it is still present in many farms, not only in Chile but also in all major Atlantic salmon producing countries, as a non-pathogenic strain (HPR0), which apparently is responsible for generating occasional outbreaks by pathogenic strains due to specific deletions in the viral genome (HPRΔ) [8].

ISAV is a single-stranded RNA virus of negative polarity, with eight segments encoding at least ten proteins [9,10]: two glycoproteins (F and HE), encoded by segments 5 and 6 form the viral capsid; the proteins PB2, PB1 and PA encoded by segments 1, 2, and 4, respectively, which are the subunits of the RNA dependent RNA polymerase (RdRP) that, together with the nucleoprotein (NP) encoded by segment 3, form the ribonucleic protein complex; additionally segment 7 encodes the non-structural protein (NSP1) and the structural (NEP) protein, and segment 8 encodes two matrix proteins (M1 and M2) [10,11]. The two surface glycoproteins, HE and F, are considered as the main virulence factors, and determine the different variants, with the non-pathogenic variant HPR0 considered as the ancestral one, whose main characteristic is a “complete” segment 6, unlike the pathogenic strains that have deletions in this segment (HPRΔ) [12,13]. 

With the aim to block viral replication, a key step in the life cycle of each virus was selected to be interfered with synthetic peptides: In IPNV, the homotrimer formed by the VP2 protein, which seems to be the building block in capsid generation, to interfere with the assembly process of newly-formed viruses. In ISAV the formation of the RNA-dependent-RNA polymerase (RdRp), requiring the assembly of three different subunits to interfere in the formation of a functional RdRP.

In this work peptides specifically designed against the aforementioned key steps for each virus were chemically synthesized and tested. To accomplish the interference of each viral cycle, in silico models of the corresponding proteins and their complexes were generated to search for fruitful interactive regions able to generate peptide sequences to be synthesized and tested. As a result, one peptide from the VP2 protein of IPNV was tested in vitro and in vivo, and for ISAV three different peptides were selected from the RdRp complex and tested in vitro, with positive results. 

## 2. Results

### 2.1. IPNV

#### 2.1.1. Bioinformatics Analysis 

Primary sequences for the IPNV polyprotein and the precursor pVP2 protein of the different genogroups (Uniprot accession P05844, Q990P9, Q703G9, Q990P7, Q990P5, Q990P8) have an identity above 80%, and in the case of the cognate birnavirus infectious bursal disease virus (IBDV) identity was above 30% (Uniprot accession P61825). A homology model was obtained through the SwissModel server using PDB ID structure 2GSY from IBDV as a template. It was necessary to model independently the loop between amino acids 110 and 117, which was performed with SPDBViewer tools [14] to achieve the construction of the mature trimeric VP2 unit. Quality assessment of the model was made through several servers (Appendix A) 

Based on the trimeric model, we were able to determine the contact regions between the units and select the sequences to be synthesized and tested; four sequences between residues 11 and 13 belonging to the C-term of pVP2 were synthesized to be tested as interfering molecules in the trimer assembly (Appendix A). Figure 1 shows the trimeric unit of VP2, with the peptides, indicated in each chain.

#### 2.1.2. Cell Association of the Peptides 

The detection of the peptide was made using the synthesized peptide with rhodamine, and the recognition with the specific antibody anti-GIM182 was done using fluorescence microscopy (Figure 2, Appendix A). Additionally, a video was made with the Z-stacks images with confocal microscopy that shows the presence of the peptide in the cells (Appendix A). Moreover, no cytotoxic effect was detected at concentrations up to two times higher than those used in the tests, as evaluated by the trypan blue method (data not shown).

#### 2.1.3. In Vitro Assays

The synthesized peptides were purified and characterized for their use in the activity assays. MALDI-TOF mass spectrometry confirmed the peptide identity and none showed any structural trend when analyzed by circular dichroism (data not shown).

The initial evaluation was performed with de novo infection of the CHSE-214 cell line with IPNV VR-299 strain with four synthetic peptides and two of them, GIM176 (YRWNLNQTALEFD) and GIM182 (TSDLPTSKAWG), significantly decreased viral replication compared with the infection control (Appendix A); however, peptide GIM182 showed higher conservation between IPNV genogroups (Appendix A) and accordingly, the subsequent assays were performed with this peptide.

To determine the concentration of the active peptide, a range from 10^−1^ to 10^−7^ M was assayed during 4 weeks and the number of IPNV viral particles generated was evaluated. The minimum concentration displaying interfering activity was set at 10^−4^ M. Concentrations higher than that did not show significant differences in the first two weeks after treatment (data not shown).

Evaluation of IPNV VP1 and VP2 by quantitative reverse transcription-polymerase chain reaction (RT-qPCR) at early times of infection, showed a decrease in the number of copies of IPNV in the cell supernatant, but not in the cell monolayer, after 48 h of infection (Figure 3). Inhibition was specifically observed when comparing the GIM182 peptides with respect to the infection control, and there was no decrease in viral load with the unrelated K1 peptide.

#### 2.1.4. In Vivo Assays

The wet laboratory test was carried out in four different stages to unify the conditions of the treatments, and it was controlled for 56 days. In this time period, mortality was recorded daily and occurred only during the first 24 days with the greatest difference between treatments observed in the first 10 days (Figure 4). The fish analyzed after the challenge was positive for IPNV, including those of the mortality control line.

The lowest mortality was recorded in those lines with the peptide, both in the control peptide line, where there was no challenge with the virus, and in the pre-challenge and post-challenge test lines (TL1: peptide added before the virus, and TL2: peptide added after the virus). The Mantel–Cox statistical analysis showed significant differences between all survival curves at a *p*-value < 0.0005. A paired analysis showed that the greatest differences occurred between the control virus line and the TL1 along with the control peptide. TL2 also showed a significant difference at a *p*-value of 0.042 (Appendix A). The three survival curves of the control peptide and the experimental lines of the treatments TL1 and TL2 lines did not show significant differences between them.

In the second in vivo assay, the treatment was applied to IPNV-carrier fish during transport to the sea. The objective was to determine if the treatment could protect persistently infected fish from an eventual outbreak, at the time of greatest susceptibility: smoltification. No mortality was observed during this assay. In this case, the IPNV VP2 expression was analyzed in relation to the elongation factor (ELF) and beta-actin genes by RT-qPCR. Figure 5 shows the results corresponding to the average of the last two samplings, the data corresponding to the first two weeks did not show any difference. The results showed no significant change in the expression of IPNV VP2 in the fish that were exposed to the peptide compared to those that were in the control tanks; although the difference was not significant, a tendency to decrease the viral load was observed (Figure 5).

In the third in vivo assay, also carried out with persistently infected fish, the relative expression of IPNV VP2 was compared between controls and samples from a pool of fry fish at 30 and 60 days after the addition of the peptide to the water. The results showed a significant decrease of IPNV VP2 relative expression in the samples from treated fish compared to samples from control fish, and such a decrease was proportional to time (Figure 6). In this assay, there was no IPNV outbreak and no mortality occurred.

### 2.2. ISAV

#### 2.2.1. Bioinformatics Analysis 

Sequences reported for the proteins PB2, PB1, and PA (encoded by segments 1, 2, and 4 respectively), obtained from NCBI presented over 96% identity with proteins belonging to the European genogroup, and above 70% identity with those of the North American genogroup, for the three subunits (Appendix A). 

The structural models for each subunit were performed with proteins from segment 1, 2 and 4 of the isolate SK779 as reference (GenBank Accession ABW93485, ABW93483, ABW93482); through ITASSER server [16], the structural assessment of the subunits was made by means of PDBSum and ProSa servers [17,18] (Appendix A). A model of the complex was made taking as reference the RdRP of influenza virus (PDB ID 4WSA, 4WRT, and 4WSB). The ISAV RdRp model is presented in Figure 7.

This model along with the information of interaction regions homology with influenza [19,20,21], was used to define the interfering peptide sequences. 

A set of three peptides of 12–15 residues, GIM535, GIM538, and GIM539, was synthesized, characterized, and purified to be tested in the fish cell lines.

#### 2.2.2. Cell Association of the Peptides and Cytotoxicity

As in IPNV, rhodamine-labeled peptides could be detected by fluorescence microscopy associated with SHK-1 cells (Figure 8), and by confocal microscopy (Appendix A). Peptides did not show the cytotoxic effect at concentrations up to two-fold higher than used in the tests (data not shown).

#### 2.2.3. In Vitro Assays

A preliminary test was performed with the peptides designed for the ISAV RdRdP. Additionally, two unrelated peptides were included: GIM182 peptide and the second one being an analog of the first (GIM1094). The tests were performed at two concentrations, 10^−3^ and 10^−4^ M, and two different times, 12 and 24 h using the HPR14a strain (Appendix A). Although the results were in general not significant (only a significant difference was observed between the GIM1094 peptide and the others at a concentration of 10^−3^ M at 24 h), the results in this assay suggested that the effect of the peptides is specific and that the concentrations used produced a decrease in the viral load. The following tests were performed only with the RdRP peptides. 

Peptides were tested at concentrations between 10^−4^ and 10^−6^ M with one ISAV strain. The results showed that the peptide was able to decrease the viral load in all cases, being this effect concentration-dependent for one of the GIM538 peptides, while the GIM539 peptide presented a similar activity at the three concentrations tested (Figure 9A). These results were observed with the specific primers directed against the genomic RNA, but not against the transcription RNA (results not shown).

In a subsequent assay, the three peptides were tested at 10^−4^ M concentration with two different ISAV strains. No results were obtained for peptide GIM539 with the HPR3a strain, due to a technical problem. The other tests showed that the peptides were able to significantly reduce the viral load for the two strains tested (Figure 9B).

## 3. Discussion

Nowadays research in peptides as therapeutic molecules has been revisited, moreover in the present situation of new virus emergence. Research on antiviral peptides (AVP) has increased thanks to the advantages that they present over other compounds, such as the variability in mechanisms of action and the low possibility of generating resistance [22]. Although the focus is primarily directed at human pathogens, aquaculture has also been considered in these studies [23]. Currently, the antiviral database AVPdb [24] compiles 2683 peptides against 67 different viruses, with 5% of them related to fish diseases.

In this report, we tested chemically synthesized peptides, designed by bioinformatics means, against the formation of the trimer of the IPNV VP2 protein and the RdRP of ISAV.

### 3.1. IPNV

The in vitro assays showed that peptide GIM182 was able to reduce the number of viral copies in at least one log compared to the control when the measurement was performed on the supernatant but not on the cells. The difference between the supernatant and the cell monolayer was, as expected, due to the technique used for virus detection (RT-qPCR) which in the cell would include all the genetic material of the virus, and also the mRNA (not necessarily complete virions), while in the supernatant mostly extracellular complete particles should de detected. However, although a significant difference was observed between the controls and treated cells, the degree of infectivity could only be determined by viral titration, as has been reported in similar works for the antiviral peptide test [25].

The experimental challenge in the wet lab yielded an unexpected result; first, the mortality control line behaved like the virus control and although all the tanks were kept in the same conditions, the fish included in the trial had a pre-existing infection with IPNV, which could have caused these mortalities; second, mortality occurred only during the first 24 days (Figure 4). Despite this, it was possible to observe significant differences between the survival curves of virus control and the fish that were exposed to the peptide. A slightly better response was observed when the peptide was added before the virus, at TL1 or pre-challenge, than when it was added after infection, or at the post-challenge condition (TL2). This behavior is similar to that observed in a study by Ojeda et al. [26] where a synthetic peptide was tested for antiviral activity against ISAV.

In the transportation tanks assay, there was no challenge with the virus and the fish included in the trial had a persistent IPNV infection. The peptide was applied to the water and the fish were in contact with the peptide for 6 h. No development of the disease occurred along with this trial and there was no record of mortality. The result of the IPNV VP2 expression analysis showed no significant difference between the fish in the tanks treated with the peptide and those in the control tanks. Despite this, the test was valuable since it allowed establishing a methodology to apply the peptide, besides verifying its safety for the fish.

The results for the third in vivo assay showed a decrease in the viral load for the fish exposed to the peptide, with the greatest effect on samples analyzed at 60 days, which suggests a cumulative effect due to the repeated application of the peptide. 

This evidences that in persistent infections the exposure to the peptide decreases the viral load, thus reducing the risk of dissemination of the infection by those fish.

The trials were not extended any further because of a lack of interest in the subject by the salmon farms, but from our point of view, it constitutes a proof of concept related to the functionality of the interfering peptides.

### 3.2. ISAV

The results show that there was a significant decrease of the virus with the use concentration of the peptides to 10^−4^ M. The activity of the peptides is theoretically due to the interference with the assembly of the three subunits of the polymerase, which in turn can interfere with the processes of viral replication or transcription. The results suggest that the interfered process is the replication, as can be seen in the reduction of the expression of the genomic RNA (Figure 9). In this context, the use of peptides for the treatment of sick fish would allow the reduction of viral load, as well as the virus spreading.

Nowadays it is known that the highest prevalence corresponds to the apathogenic variant HPR0, which is present in practically all countries that grow Atlantic Salmon [27], and sporadically (2 to 3 cases with HPRΔ per year in Chile) due to deleterious variants that cause the disease. The current regulations in Chile do not allow the treatment of animals infected with pathogenic variants of the virus and sanitary slaughter is enforced; however, animals infected with apathogenic variants (HPR0) can continue in the production cycle under intensive surveillance being aware that the situation can change at any time [8,28]. The results obtained in our study, show that the peptides can decrease the viral load of the variants tested (HPR3a and HPR14a). Considering that they are directed against the RdRdp, which is a highly conserved protein among the different virus variants (including the non-pathogenic HPR0), there is then a high probability that the peptides could be also applied against HPR0 variant. However, up to now, it has not been possible to cultivate the HPR0 strain in the laboratory, therefore, it seems appropriate to carry out in vivo tests with fish infected with this variant to confirm it.

This is another scenario in which the peptides could be used against the non-pathogenic variant of the virus, thus eliminating the risk of the appearance of virulent variants, which can be generated from them. Its application in periods where the elimination of fish entails a great economic loss, as it is in breeding fish with high genetic value, or in the final stages of the fattening phase, could be an important contribution to the aquaculture industry.

## 4. Materials and Methods 

### 4.1. Bioinformatics Analysis and Homology Modeling

NCBI and Uniprot database were used to obtain the sequences, and secondary structure prediction was performed through the Jalview platform [29], and the CLC Main workbench package (Qiagen bioinformatics).

#### 4.1.1. IPNV

Sequences from six of the IPNV genogroups and two sequences of IBDV were compared by pairwise and multiple alignments with the Clustal algorithm [30].

The VP2 sequence of IPNV Jasper strain (Uniprot accession number P05844) was used to construct a homology model of the 3D structure of the protein. At the time when the process began the experimental structure of IPNV was not available.

IBDV tridimensional structures of the capside were determined by Garriga et al. (PDB ID: 1WCD, 1WCE) [24,31]; these structures were used to generate the model of IPNV VP2 homotrimer and, subsequently, the structure reported for IPNV by Coulibaly et al. (PDB ID 3IDE, Coulibay et al. 2010) allowed to confirm the capsid composition and detail the interaction established in the VP2 homotrimers.

The model was made through SwissModel Server [32] analyses and visualization was made using SwissPdb Viewer [14] and Chimera [15]. 

Structural assessment of the model was made through PDBSum and Prosa Servers [17,18] 

#### 4.1.2. ISAV

Sequences obtained for each RdRp subunit (PB2, PB1, and PA) were aligned in CLC Main workbench, and the identity percentage was determined by pairwise comparison. 

In addition, each subunit was compared with their counterpart in influenza, and the interaction regions were established by homology with ISAV.

Sequences of the reference isolate SK779, with Genbank accession ABW93482.1, ABW93483.1, and ABW93485.1for the three subunits PB2, PB1, and PA, respectively, were used to construct the 3D Model, through ITASSER server [16].

Each subunit model was subjected to structural assessment through PDBsum and Prosa servers [17,18].

### 4.2. Peptide Synthesis, Purification, and Characterization

Selected sequences were synthesized by solid-phase multiple peptide system on Rink amide resin (0.65 meq/g) using Fmoc amino acids (Iris Biotech). Moreover, a portion of peptide-resin was used for Rhodamine B coupling (for subsequent use in microscopy). 

The peptide cleavage was performed with a solution of TFA/TIS/EDT/H_2_0 (92.5% trifluoroacetic acid /2.5% triisopropylsilane /2.5% 1.2-ethandithiol /2.5% ultrapure water), washed with cold ether and finally purified by RPHPLC to a purity higher than 95% with a 0–70% acetonitrile-water mixture gradient over 30 min at a flow rate of 1 mL/min. The peptides were characterized by HPLC and mass spectrometry and purified by C18 columns before use. The secondary structure trend of the peptides was determined by circular dichroism (CD) at 25 °C between 190–260 nm on a JASCO J-815 CD Spectrometer (Jasco Corp., Tokyo, Japan) as described before [33]. 

Peptide identity was confirmed by MALDI-TOF mass spectrometry. For that purpose 1 μL of 1 μg/μL peptide solution along with 1 μL of matrix (10 mg/mL of α -cyano-4-hydroxycinnamic acid in 50% acetonitrile with 0.1% formic acid in water) was loaded onto a micro scout sample plate (Bruker Daltonics Inc., Billerica, MA, USA) and the measures were made in a MALDI-TOF-MS Microflex equipment (Bruker Daltonics Inc.) using flexControl 3.0 software (Bruker Daltonik GmbH, Bremen, Germany). Spectra were obtained in positive ion mode by reflection detection as additions of 10 rounds of 30 laser impacts each, at various points of the sample.

### 4.3. Cell Lines and Virus Strains

#### 4.3.1. IPNV

Common blue gill embryo cells (CHSE/F) (formerly known as CHSE-214, ATCC CRL-1681) [34] and CHSE/F cells persistently infected with IPNV-NVI015, were cultured in Leibovitz media (L-15, Gibco), supplemented with 10% (v/v) fetal bovine serum (FBS, Gibco) and 2 mm glutamine (Gibco) and maintained as previously described [35]. 

The IPNV strain used was the Chilean strain VR-299 obtained from a natural outbreak in the lake district in Chile (X Region). CHSE/F cells monolayer were infected at semi-confluency at a multiplicity of infection of 0.001 in Leibovitz media (L-15, Gibco) at 18 °C, supplemented with 50 µM gentamicin, 2 mM L-glutamine and 10% fetal bovine serum (FBS, Gibco). Once full cytopathic effect was observed, culture fluids were harvested and tittered, quantifying the number of infective particles, using a modified Reed and Muench protocol [36].

#### 4.3.2. ISAV

Atlantic salmon head kidney cells (SHK-1) cells (ATCC CRL-2747) [37] were cultured in Leibovitz media (L-15, Gibco) supplemented with 10% (v/v) fetal calf serum (FCS) and 4mM glutamine (Gibco) at 20 °C. 

ISAV strains correspond to field isolates, HPR3a and HPR14a types, maintained in the Laboratorio de Genética e Inmunología Molecular collection. Virus propagation is performed using SHK-1 cells: 80% confluent cells monolayer were incubated for four hours with a virus strain dilution in L-15 media; after that cells were washed twice with fresh media, and cultured with L-15 media supplemented with FBS 2% and antibiotics, at 17 °C for 7 days. The virus is then recovered from the supernatant and filtered through 0.45 μm, to aliquot and the virus was stored at −80 °C. Virus titration is performed by a plaque assay, as previously described [38].

### 4.4. Cell Association of the Peptides 

Peptides labeled with rhodamine were tested for their capacity to enter the cells. Briefly, exponentially growing cells of Chinook salmon embryo (cells line CHSE-214) were exposed to 100 ng/well concentration of rhodamine-labeled peptide for 3 h. For direct detection of rhodamine-labeled peptides, the culture medium was discarded, and the cells were washed once with phosphate buffer saline (PBS) (pH 7.3), subsequently rinsed three times with PBS (pH 7.3) plus Tween 20 (0.05%) and viewed under a Nikon Eclipse 400 fluorescence microscope. Colour photography was performed with a Nikon Coolpix 4500 digital camera. Additionally, the production of a polyclonal antibody was requested from the company Pickcell^®^ (The Netherlands), obtained in the rabbit, and used to check the cell association of the peptide GIM182 without rhodamine into the CHSE-214 cells.

### 4.5. Peptide Effect over Cell Lines Infected with IPNV

CHSE-214 cells and IPNV persistently infected cells were used to evaluate the peptide effect.

For de novo infection CHSE-214 cells were grown on 24-well cell culture plates to confluence and infected with IPNV at a multiplicity of infection of one (MOI = 1); at 18 h post-infection, the cells were washed and each peptide added at 10 µM during four hours, washed and then fresh medium added.

CHSE-214 cells persistently infected with IPNV-NVI015 were grown on 24-well until confluence [39]; the peptide was added at a total concentration of 10 µM for four hours, washed and then fresh medium was added; measurements were made in triplicate. 

After the peptide treatment, total RNA was extracted, from the supernatant and cell monolayer, by using TRizol^®^ reagent (Life Technologies, Carlsbad, California, USA). IPNV expression was measured by VP1 and VP2 amplification using RT-qPCR real-time procedure, and ELF-1α as housekeeping gene (Stratagene 1 step RT-qPCR Kit) with specific primers for each segment (Appendix A). The reaction was carried out in a 30 µL mixture consisting of Brilliant^®^ II SYBR^®^ Green QRT-PCR Master Mix Kit, 1-Step (Stratagene, Inc.), primers and template RNA, as indicated by the manufacturer. The samples were amplified and detected using a Chromo 4 system (BioRad). The final thermocycling profile was 50 °C for 55 min, 94 °C for 10 min and 94 °C for 30 s, 55 °C for 30 s and 72 °C for 40 cycles. 800 ng of total RNA per sample were used in each triplicate reaction. To quantify VP2 mRNA copies of IPNV, a standard curve for DNA quantification was established. To have IPNV real-time PCR standards, the VP2 region was amplified using published procedures standardized in our laboratory [40]. 

All the experiments included appropriate controls: positive control infections without peptide, negative controls with all the reagents but peptide and virus (only in the case of de novo infection), and a non-related peptide at the same concentrations of the IPNV peptides (peptide K-1 from *Trypanosoma cruzi* [41]).

### 4.6. Peptide Effect over Cell Lines Infected with ISAV

To evaluate the effect of the peptides on the RdRp of ISAV, extraction of the total RNA from SHK-1 cells infected, in presence and absence of each peptide, was carried out by using the TRIzol^®^ procedure, and specific primers were used to evaluate each of the RNAs generated in the life cycle of the virus (vRNA, cRNA, mRNA). The assay was performed on 24-well cell culture plates. To determine the minimum active concentration each peptide was initially tested at concentrations of 10^−4^, 10^−5^ and 10^−6^ M. The cells were preincubated with the peptide for 3 h, then washed with 1X PBS, and infected with 250 µL of ISAV (HPR14a strain) with a viral titer of 10^5^ TCID50 mL^-1^. The cells were then washed with 1X PBS and fresh L15 medium with antibiotics and 2% fetal bovine serum added and incubated for 12 h. Then, the supernatant in each well was collected and stored at −80 °C for subsequent analysis. Lysis buffer was added to the floor of each well, and the total RNA from the infected cells and controls was extracted by using the EZNA^®^ Total RNA Kit I. For the analysis, segment 8 of the virus was specifically amplified; in addition, the elongation factor ELF-1α was amplified to evaluate the quality of the extracted material. The cDNA synthesis of each sample was performed using M-MLV retro-transcriptase enzyme (Promega) at 37 °C according to the supplier’s recommendations. The products were detected by real-time PCR, using the Brilliant III ultra-fast qPCR Master-Mix kit (Stratagene). The PCR reaction was carried out in 20 µL final volume using: 100 ng of total cDNA, 10 µL of 2X SYBR Green Q-PCR master mix, and 0.6 µL of each primer at 250 nM final concentration. The amplification protocol comprised a denaturation cycle at 95 °C for 3 min, followed by 40 cycles with 5 s at 95 °C and 10 s at 60 °C.

A second assay was performed using two different ISAV strains (HPR3a and HPR14a) under the same conditions and a peptide concentration of 10^−4^ M, all the measures were made in triplicate. 

The cycle threshold (Ct) of each sample was evaluated, using forward primers specific for vRNA [42], together with elongation factor (ELF) as housekeeping gene (Appendix A).

### 4.7. In Vivo Assays

#### 4.7.1. Wet Lab Assay

The assay was carried out in the Aquaculture Engineering Laboratory of Universidad Católica de la Santísima Concepción (UCSC). For the purpose, 5 circular plastic tanks with 250 5 g *Salmo salar* fry were used. The tanks are of adjustable water volume, which allows modifying the density from 25 kg/m^3^ to 50 kg/m^3^. The recirculation rate of water was 4 times per hour, with a change rate of 15 cm^3^ per sec. The average temperature was 11.5 ºC. The VR-299 strain of the IPNV was used at a concentration of 10^5^ viral particles per mL, and the peptide concentration used was 10^−5^ M. Two treatments in four different stages were tested, with stages 2 and 4 being common for all tanks, and three types of controls were used according to the distribution shown in Table 1.

Treatment 1 where the peptide is added before the virus was called pre-challenge and treatment 2 was called post-challenge.

The total trial lasted 67 days, 11 days of acclimatization, and 56 days of evaluation. Mortality was recorded daily for all the tanks used in the assay.

#### 4.7.2. Transportation Tank Assay

An initial group of 300,000 20 g *Salmo salar* pre-smolt were transferred between two farming centers. The transport was carried out in two groups (150,000 each) at a density of 40 kg/m^3^ and a temperature of 8–12 °C, in cubic fiberglass reinforced plastic transport tanks of 3 m^3^ (50 tanks in total), with oxygen supplementation. Prior to the transfer, 84 of these fish, randomly sampled from the initial group, were analyzed to determine the presence of IPNV by RT-qPCR applied to RNA extracted from fish kidney and gills.

During the transfer, peptide GIM182 was added to the water to one of the groups at a concentration of 10^-7^ M and the other group was kept as control. The transfer time and, therefore, the exposure time of the pre-smolt fish to the peptide, was 6 h.

Upon arrival, the fish were placed in four 80 m^3^ circular tanks of fiberglass reinforced plastic of, with a flow of 90 L/min and a rate of change of 15%, and an average temperature of 11.5 °C. Two of the tanks were used for the fish treated with the peptide (TK3 and TK7) and the remaining two used as controls (TK5 and TK8). The experiment was maintained and kept at these conditions (density of 20 kg/m^3^) for 4 weeks.

For each tank of treated and untreated fish, samples were taken once a week, as scheduled in Table 2. For sampling, the kidneys and gills were removed and stored in RNAlater^®^; afterward, they were processed with TRIzol^®^ for RNA extraction.

#### 4.7.3. IPNV Persistently Infected Fish Assay

A final assay in collaboration with a salmon company was carried out in a freshwater facility using fish with a persistent IPNV infection. A number of 1.1 million *Salmo salar* fingerlings of 0.5 gr were used, distributed in 33 circular fiberglass reinforced plastic tanks with a density of 15,000 fish per cubic meter: 7 tanks used as controls, and 26 for treatment with the peptide.

The treatment was carried out on fish cultured in open flow in its first stage and then entered into a recirculation system.

The capacity of the tanks employed was of 2 m^3^, which was reduced to 1.6 m^3^ during the application of the peptide, with a final concentration of peptide of 1.6 × 10^−7^ M. Baths were carried out on days 6, 9, and 12 and samples were taken from 30 fish on day 30 and day 60 for IPNV analysis.

The treatment with the peptide was carried out by bath on days 6, 9, and 12, and samples were taken distributed as follows:Three groups from the control tank at 0, 30, and 60 days. 5 fish for each group.Fish from the treatment tank at 30 days. 5 fish.Fish from the treatment tank at 60 days. 5 fish.

The analysis was made using the whole fish, placing it in a sterile bag, macerated with a rubber hammer, until obtaining a homogeneous mixture. Aliquots of 300 μL were taken from this mixture and RNA extraction was carried out by the TRIzol^®^ method. The RNA was quantified and the amplification of VP2 was followed as described for in vitro test. The expression of VP2 with respect to the ELF host gene was assessed by using the 2-ΔΔCT method, as described by Livak and Schimttgen [31].

### 4.8. Statistical Analysis

Graphics and statistical calculations were performed by the Graphpad Prism v6.1 software (La Jolla, CA, USA, www.graphpad.com). Results were expressed as mean plus standard deviation and analyzed by one-way or two-way ANOVA followed by Dunnett’s multiple comparisons. Survival curves were analyzed with the Long-rank Mantel–Cox test.

### 4.9. Ethics Statement

In vivo assays and sampling were carried out for veterinarians in accordance with the standards and norms defined by the responsible entity in Chile, SERNAPESCA, and in compliance with the procedures and standards established at “The Aquatic Animal Health Code” chapter 7 of the World Organization for Animal Health (OIE) [43].

## 5. Conclusions

In recent years significant progress has been made to elucidate the mechanisms of action of antimicrobial peptides (AMPs) and their involvement in disease control in humans [44]; notwithstanding, there are other biological systems in which this approach should be highly beneficial. One of them is aquaculture, where confinement is a challenging threat that opportunistic pathogens dare to take advantage of. Among different pathogens, viruses appear to be the most elusive ones with a high capacity of adaptation. In salmon production, RNA viruses appear to be one of the hidden enemies with well-defined strategies to persist and therefore immunologically weakening their targets. This is mainly because fish lack a powerful adaptive immune response, thus questioning the use of vaccines as preventives strategies, and in general, the immune response displayed is not enough to control viral infections.

IPNV and ISAV are certainly good examples, since curing therapies do not exist for either of them, although the primary mechanism of AMP-mediated antiviral activity has been attributed to direct interference with viral envelopes, which should apply to the case of ISAV reported here. Nonetheless, the also effective antiviral activity against non-enveloped viruses, such as IPNV, hints at the presence of undiscovered activities of AMPs perhaps associated to post-entry steps and/or affecting later stages in the viral life cycle, such as genome replication and viral protein trafficking. Additionally, an optimistic outcome from the fact that AMP exposure prior to viral infection results in peptide retention and internalization by the cells, which may reflect a more robust response to viruses compared to other potential therapeutics in development. Finally, the simplicity of AMPs makes the development of synthetic peptide analogs a cost-effective strategy to treat established viral infections. Synthetic peptides are versatile molecules that can be used in many different scenarios, from basic investigation to technology and development, becoming powerful tools of fresh air in the pharmaceutical market.

In this work, we presented the positive effect of peptides directed against two different types of RNA virus with two different targets, but in either case, interfering in crucial steps of viral development. In summary, we presented evidence that the use of peptides as antiviral agents in disease control may be a viable alternative to explore in aquaculture.

## 6. Patents

IPNV peptide is included in the patent PCT/IB2014/063047. “Agent with antiviral properties for preventing or treating individuals exposed to a virus of the *Birnaviridae* Family”.

## Figures and Tables

**Figure 1 pathogens-09-00600-f001:**
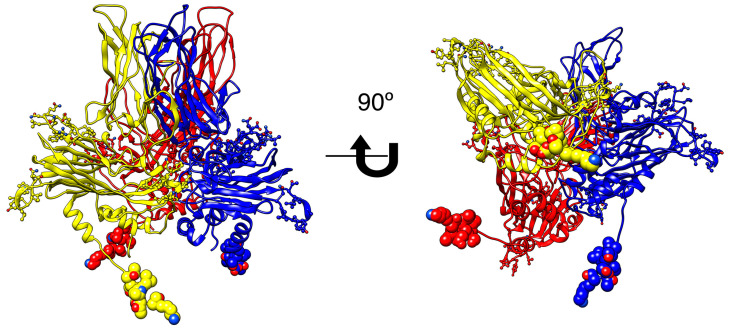
Homotrimeric model for the pVP2 precursor protein of the Jasper strain of infectious pancreatic necrosis virus (IPNV) (P05844 Uniprot accession). Each chain in ribbon representation has a different color; GIM182 peptide is indicated in spheres representation, the other peptides are indicated in ball and stick. The model was constructed through the SwissModel server and analyzed with Chimera [15].

**Figure 2 pathogens-09-00600-f002:**
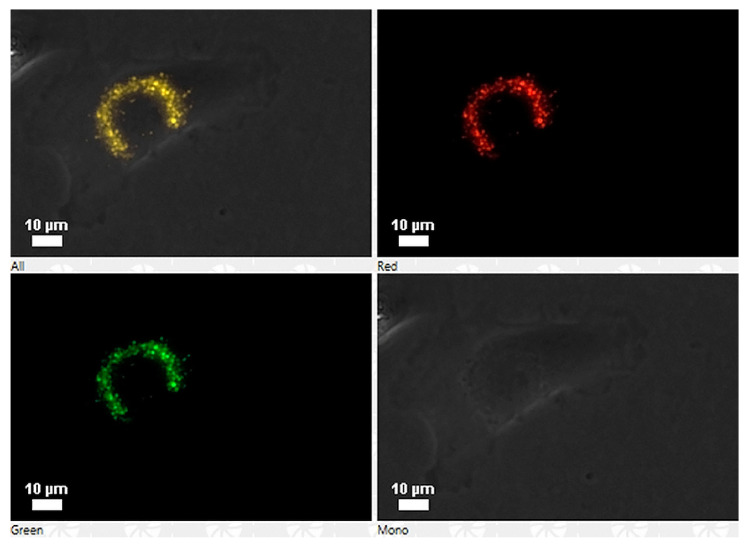
Association of the peptide to the cell revealed by fluorescence microscopy. In red the peptide rhodamine, in green the peptide recognized by a specific antibody, the bright field is shown, and the merge of the three layers is in yellow.

**Figure 3 pathogens-09-00600-f003:**
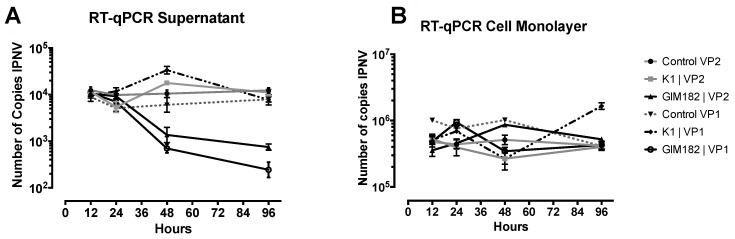
Evaluation by RT-qPCR of copy number of IPNV in supernatant (**A**) and cell monolayers (**B**): determination of VP2 and VP1 copies. Control VP2 or VP1: positive control IPNV without peptide; K1: peptide non-related, GIM182: IPNV with peptide at 10^−4^ M. The measurements were made in triplicate and the values are represented with their standard deviation.

**Figure 4 pathogens-09-00600-f004:**
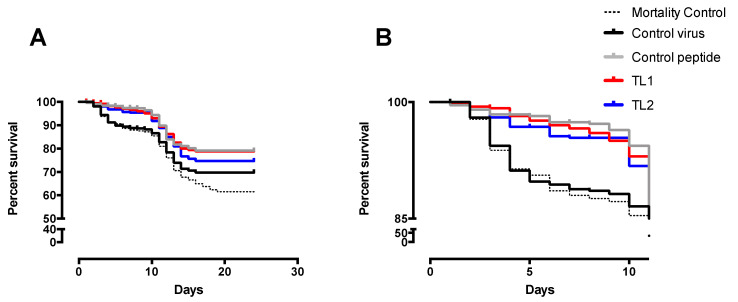
Survival curves from the wet lab assay. (**A**). TL1: test line 1 with peptide GIM182 added at a 10^−5^ M concentration before the virus challenge; TL2: test line 2 with peptide GIM182 added after the virus challenge; control virus: virus without peptide addition; control peptide: peptide without virus; mortality control: only water. (**B**). Extract of the survival curves for the first ten days of the assay.

**Figure 5 pathogens-09-00600-f005:**
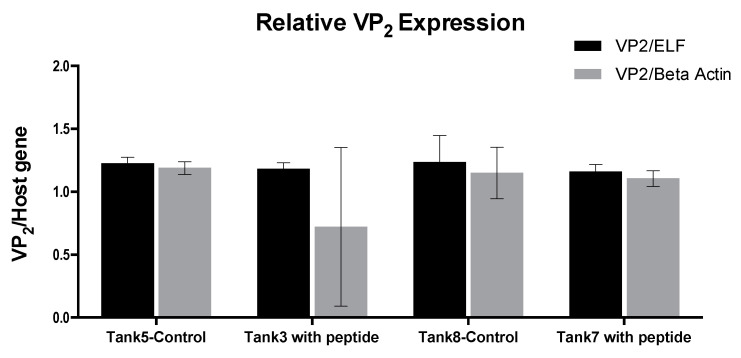
IPNV VP2 gene expression level of by RT-qPCR, on RNA extracted from fish kidney and gills. The comparison was made with two host genes as reference: elongation factor (ELF) and beta-actin. The peptide was added at a 10^−7^ M concentration. Error bars represent the standard deviation.

**Figure 6 pathogens-09-00600-f006:**
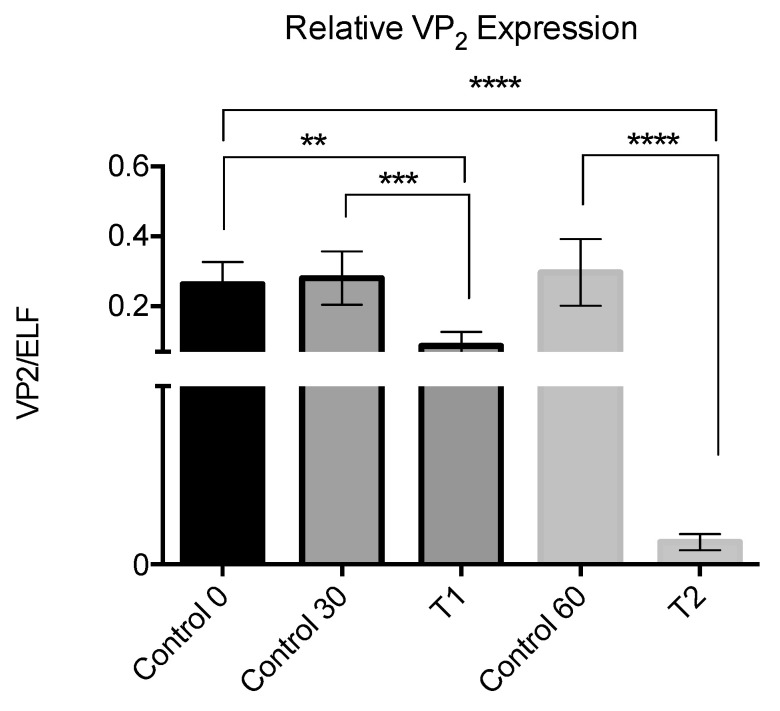
IPNV VP2 gene expression level in comparison with ELF as a host gene by RT-qPCR on RNA extracted from the whole fish. The peptide was added at a 10^−7^ M concentration in days 6, 9, and 12; controls and samples were taken at two different times T1: 30 days, and T2: 60 days. Results were significant with respect to the control at *p* < 0.0001 (****), *p* < 0.001 (***) and *p* < 0.05 (**).

**Figure 7 pathogens-09-00600-f007:**
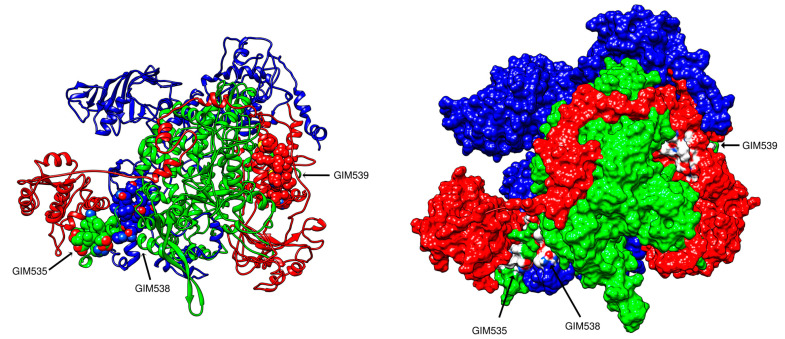
A structural model for the RdRp complex formed by the three subunits PB2 (blue), PB1 (green), and PA (red). The peptides are indicated in the model.

**Figure 8 pathogens-09-00600-f008:**
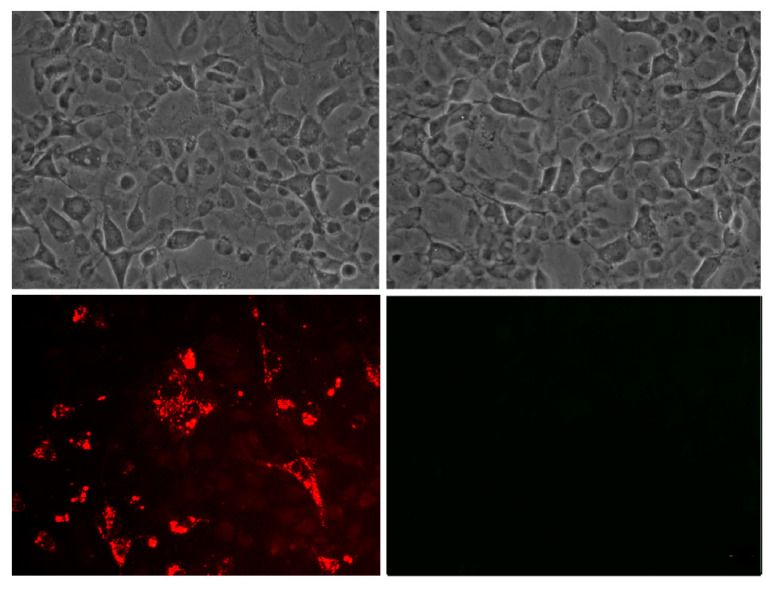
Association of peptide GIM538 with cells SHK-1 by fluorescence microscopy. The left panel shows the bright field and the peptide with rhodamine (left/bottom), and the right panel the peptide without rhodamine (right/bottom).

**Figure 9 pathogens-09-00600-f009:**
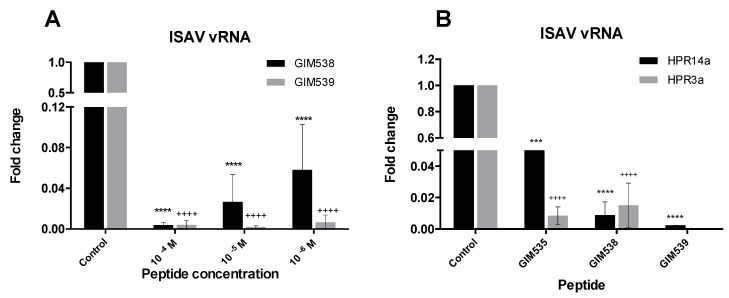
In-vitro assays of peptides against ISAV RdRp. (**A**): minimum active concentration determination (HPR14a strain was used). (**B**): activity of the peptides against the two ISAV strains; the peptides were added at a 10^−4^ M concentration. Significant differences between control and each peptide are indicated with ****, or ++++ *p* < 0.0001, and *** *p* < 0.001. Error bars represent the standard deviation of triplicate measurements.

**Table 1 pathogens-09-00600-t001:** Distribution of the treatments for the wet lab assay.

Stage	Treatment 1 Tank 1	Treatment 2 Tank 2	Control Virus Tank 3	Control Peptide Tank 4	Mortality Control Tank 5
1	**C2**** Peptide X 4 h	**C2** IPNV X 3 h	**C2** IPNV X 3 h	**C2** Peptide X 4 h	**C2** X 4 h
2	**C1*** X 1 h				
3	**C2** IPNV X 3 h	**C2** Peptide X 4 h	**C2** X 4 h	**C2** X 3 h	
4	**C1** X 56 days				

**C1***: Culture density 25 kg/m^3^; **C2****: Culture density 50 kg/m^3^.

**Table 2 pathogens-09-00600-t002:** The number of fish sampled weekly in treated and control tanks.

Tank/Treatment	Week 1	Week 2	Week 3	Week 4	Total
TK3/Peptide	50	50	40	50	190
TK5/Control	50	50	40	22	162
TK7/Peptide	50	46	40	50	186
TK8/Control	50	40	40	50	180

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
