# Peer review of "Synthetic Peptides as a Promising Alternative to Control Viral Infections in Atlantic Salmon"

_pathogens, 2020, doi:10.3390/pathogens9080600_

Round 1
Reviewer 1 Report
What do you want to do ? New mailCopy
The manuscript “Interfering synthetic peptides as a novel tool to control viral infections in fish”, by Carden et al, to be published in Pathogens, is potentially an important contribution to science in general, and more precisely to the control of a couple of important viral pathogens, IPNV and ISAV. The experimental design has been conscientious, extensive and perfectly adapted to the established objective, as is to be expected from this research group, of which I have already had the pleasure of reviewing some other manuscripts. For this reason, I was surprised by the quality of the manuscript, which does not merit the intense work done in the study.
Therefore, I encourage the authors not to be discouraged by this apparent rejection to their job, because it does not refer to their study, but only to the way in which it has been presented in this version. And thus, apply a deep re-editing of the manuscript, considering the suggestions I comment below, and are color-marked in the attached pdf:
GENERAL COMMENT:
Not only the language but also the writing need a deep improvement.
INTRODUCTION
General comment to this chapter: This chapter needs a deep re-edition
Com 1- Line 53: The sentence “The disease occurs mainly in fresh water, affecting fish in the fry, pre-smolt and smolt stages of development, when immunological response of fish is depressed, or in the brood stock fish with the risk of vertical transmission [4,5]” seems to be out of place. I believe it would be better inserted in line 42, before "Although it does not ...".
Com 2- Lines 39 and 57: Genus and Family names must be written in italics and with the initial in Uppercase. The first could be a mistake (line 39), but the second denotes a conceptual error. Please, correct
Com 3- Lines 65-69: The sentence “Structurally …” must be re-written.
Com 4-Line 76: "enclose" is not an appropriate word for a life cycle step. In addition, ALL VIRUSES have in the replication cycle important steps associated with assembly. This seems not to be the best way to introduce the justification of the aim of this study: There is a gap in between the previous paragraphs and this one. I would suggest to completely re-write this chapter, taken more care to the language, but mainly introducing the real subject, which is the blocking of viral replication.
Com 5.- Line 80: “we propose to test” I guess what the authors mean is that they have tested ... and propose the use of ... I do not think the aim of this study was to propose the testing of [substances] for ...
RESULTS
General comment to this chapter: After line 121, I decided not to continue reviewing this chapter since it seems that similar comments must be applied throughout all of it.
Com 1-Line92: “have an identity above 80%, and above 30%” I have changed it by “have an identity above 80 and 30% respectively”. Is it correct? Otherwise, improve the sentence making it clearer.
Com 2-Lines 98-99 and Spp Fig S1: The figure does not seem to show what is indicated in the sentence.
Com 3-Lines 106-107: In general, the figures are not explained, just the final interpretation by the authors is indicated in the text, what does not help to understand their interpretation.
Com 4-Line 111: MALDI-TOF mass spectrometry been not described in M&M
Com 5-Lines 120-121: Again, Fig 2 is not described to help getting to the same conclusion the author affirms to have reached. In addition, where is a description of the codes used for each line?
DISCUSSION
General Com: a partial description of results is included in this chapter, but it should have been in Results. Most importantly: To my understanding this is a very poor discussion for an intense and extensive (in terms of the experimental approach) study. A Discussion chapter must describe interpretation of all results, positive or negative, and their comparison with other studies, not only from the authors, but from any other groups.
Com 1-Line 204: Does the authors mean that the virus has been removed? I think it is a little risky to affirm that; it really seems that the virus has been someway blocked, but ¿eliminated?
Com 2-Line 202: Explain why is this your interpretation
M&M
General Com: I have not reviewed it all, since I expect the authors will get into a deep re-edition of the whole manuscript.
Com 1-Line 296: Table S2 does not show the specific primers for each segment!
Com 2-Line 105: Did I miss a list of peptides? It would be useful to, for example, be able to interpret figure 2.
What do you want to do ? New mailCopy What do you want to do ? New mailCopy
Author Response
First of all we want to thank the reviewers for their extensive review and enriching comments.
We also want to apologize for the mistakes and oversights, and we hope that the current version of the manuscript complies with the requested revisions.
Below is a detailed list of changes made according to each of the reviewers.
Reviewer 1.
The manuscript “Interfering synthetic peptides as a novel tool to control viral infections in fish”, by Carden et al, to be published in Pathogens, is potentially an important contribution to science in general, and more precisely to the control of a couple of important viral pathogens, IPNV and ISAV. The experimental design has been conscientious, extensive and perfectly adapted to the established objective, as is to be expected from this research group, of which I have already had the pleasure of reviewing some other manuscripts. For this reason, I was surprised by the quality of the manuscript, which does not merit the intense work done in the study.
Therefore, I encourage the authors not to be discouraged by this apparent rejection to their job, because it does not refer to their study, but only to the way in which it has been presented in this version. And thus, apply a deep re-editing of the manuscript, considering the suggestions I comment below, and are color-marked in the attached pdf:
GENERAL COMMENT:
Not only the language but also the writing need a deep improvement.
INTRODUCTION
General comment to this chapter: This chapter needs a deep re-edition
Com 1- Line 53: The sentence “The disease occurs mainly in fresh water, affecting fish in the fry, pre-smolt and smolt stages of development, when immunological response of fish is depressed, or in the brood stock fish with the risk of vertical transmission [4,5]” seems to be out of place. I believe it would be better inserted in line 42, before "Although it does not ...".
R/ Done
Com 2- Lines 39 and 57: Genus and Family names must be written in italics and with the initial in Uppercase. The first could be a mistake (line 39), but the second denotes a conceptual error. Please, correct
R/ Corrected
Com 3- Lines 65-69: The sentence “Structurally …” must be re-written.
R/ It was rewritten as follows:
L68-72
ISAV is a single stranded RNA virus of negative polarity, with eight segments encoding at least ten proteins [8,9]: two glycoproteins (F and HE) encoded by segments 5 and 6 form the viral capsid, the proteins PB2, PB1 and PA encoded by segments 1, 2 and 4 respectively, which are subunits of the RNA dependent RNA polymerase (RdRP) that, together with the nucleoprotein (NP) encoded by segment 3, form the ribonucleic protein complex.
Com 4-Line 76: "enclose" is not an appropriate word for a life cycle step. In addition, ALL VIRUSES have in the replication cycle important steps associated with assembly. This seems not to be the best way to introduce the justification of the aim of this study: There is a gap in between the previous paragraphs and this one. I would suggest to completely re-write this chapter, taken more care to the language, but mainly introducing the real subject, which is the blocking of viral replication.
Com 5.- Line 80: “we propose to test” I guess what the authors mean is that they have tested ... and propose the use of ... I do not think the aim of this study was to propose the testing of [substances] for ...
R/Thanks for the comment, we changed the paragraphs as follows:
L78-88
With the aim to block viral replication, a key step in the life cycle of each virus was selected to be interfered with synthetic peptides: In IPNV, the homotrimer formed by VP2 protein as the initial step in capsid generation to interfere with the assembly process of newly-formed viruses; in ISAV the formation of the RNA-dependent-RNA polymerase (RdRp), requiring the assembly of three different subunits to interfere in the formation of a functional RdRP.
In this work synthetic peptides directed against the aforementioned key steps for each virus were tested, specifically designed and chemically synthesized. In order to accomplish these goals, in silico models were generated of the corresponding proteins and their complexes, to search for fruitful interactive regions able to generate peptide sequences to be synthesized and tested. As a result, one peptide from the VP2 protein of IPNV was tested in vitro and in vivo, and for ISAV three different peptides were selected from the RdRp complex and tested in vitro with positive results.
RESULTS
General comment to this chapter: After line 121, I decided not to continue reviewing this chapter since it seems that similar comments must be applied throughout all of it.
Thank you for bringing this to our attention, the results section was modified ad the figures described in the text.
Com 1-Line92: “have an identity above 80%, and above 30%” I have changed it by “have an identity above 80 and 30% respectively”. Is it correct? Otherwise, improve the sentence making it clearer.
R/ Sentence was modified
L92-94
Primary sequences for the IPNV polyprotein, and the precursor pVP2 protein of the different genogroups (Uniprot accession P05844, Q990P9, Q703G9, Q990P7, Q990P5, Q990P8) have an identity above 80%, and in the case of the cognate birnavirus IBDV identity was above 30% (Uniprot accession P61825).
Com 2-Lines 98-99 and Spp Fig S1: The figure does not seem to show what is indicated in the sentence.
R/ Figure S1 was changed according to the sentence. L101-103
Com 3-Lines 106-107: In general, the figures are not explained, just the final interpretation by the authors is indicated in the text, what does not help to understand their interpretation.
L111-116
The detection of the peptide was made using the synthesized peptide with rhodamine, and the recognition with the specific antibody anti-GIM182 was done using fluorescence microscopy (Figure 2 , supplementary Figure S2). Additionally, a video was made with the Z-stacks images with confocal microscopy that shows the presence of the peptide in the cells (supplementary video GIM-182).
Com 4-Line 111: MALDI-TOF mass spectrometry been not described in M&M
R/ It is now included in Methods.
L355-361
Peptide identity was confirmed by MALDI-TOF mass spectrometry. 1mL of 1mg/mL peptide solution along with 1mL of matrix (10 mg/mL of a -cyano-4-hydroxycinnamic acid in 50% acetonitrile with 0.1% formic acid in water) was loaded onto a micro scout sample plate (Bruker Daltonics Inc., Billerica, MA, USA) and the measures were made in a MALDI-TOF-MS Microflex equipment (Bruker Daltonics Inc.) using flexControl 3.0 software (Bruker Daltonik GmbH, Bremen, Germany). Spectra were obtained in positive ion mode by reflection detection as additions of 10 rounds of 30 laser impacts each, at various points of the sample.
Com 5-Lines 120-121: Again, Fig 2 is not described to help getting to the same conclusion the author affirms to have reached. In addition, where is a description of the codes used for each line?
R/ Figure was changed, now it is figure 3, and is explained in the text.
L134-137
Evaluation of IPNV VP1 and VP2 by RT-PCR at early times of infection showed a decrease in the number of copies of IPNV in the cell supernatant, but not in the cell monolayer, after 48 hours of infection (Figure 3). Inhibition was specifically observed when comparing the GIM182 peptide with respect to infection control, and there was no decrease in viral load with the unrelated K1 peptide.
DISCUSSION
General Com: a partial description of results is included in this chapter, but it should have been in Results. Most importantly: To my understanding this is a very poor discussion for an intense and extensive (in terms of the experimental approach) study. A Discussion chapter must describe interpretation of all results, positive or negative, and their comparison with other studies, not only from the authors, but from any other groups.
The whole discussion section was rewritten and improved.
Com 1-Line 204: Does the authors mean that the virus has been removed? I think it is a little risky to affirm that; it really seems that the virus has been someway blocked, but ¿eliminated?
R/ The reviewer is right: to check if the virus has been removed additional tests are needed. We are now referring to a decrease in viral load, which is what is observed.
L281-282
This is evidence that in persistent infections the exposure to the peptide decreases the viral load, so reducing the risk of infection dissemination by these fish.
Com 2-Line 202: Explain why is this your interpretation
R/All the paragraph was rewritten
What is observed is now as follows:
L277-280
In addition, a greater decrease in the viral load was observed in the samples analyzed at 60 days, which suggests a cumulative effect of the peptide since, unlike than in the previous tests, a repeated application was carried out three times.
M&M
General Com: I have not reviewed it all, since I expect the authors will get into a deep re-edition of the whole manuscript.
Com 1-Line 296: Table S2 does not show the specific primers for each segment!
R/ Sorry this is now included in Table S4.
Com 2-Line 105: Did I miss a list of peptides? It would be useful to, for example, be able to interpret figure 2.
R/ In principle line 105 and figure 2 mentioned by the reviewer involved only the peptide GIM182, but some preliminary tests that included other peptides, are now incorporated in the text.
L126-129 and Figure S3
The initial evaluation was performed with de novo infection of the CHSE-214 cell line with IPNV with four synthetic peptides and only one of them, GIM182 (TSDLPTSKAWG), significantly decreased viral replication compared with the infection control (Supplementary Figure S3); accordingly, the subsequent assays were performed with this peptide.
Reviewer 2 Report
These experiments present interesting results from the use of peptides to decrease viral loads. If the efficacy of these peptides can be independently confirmed, they have the potential to provide value to the aquaculture industry. Indeed, in some areas of the world where ISAV is endemic, regulatory agencies do not automatically force ISAV-infected farms to depopulate (e.g., eastern Canada).
I offer several suggestions for revision to improve clarity of the presentation.
Abstract – Some of the recommended components of a Pathogens abstract are not clear here (see https://www.mdpi.com/journal/pathogens/instructions):
“Background: Place the question addressed in a broad context and highlight the purpose of the study” – Not clear to me in the abstract are the question addressed or the purpose of the study.
Methods: what species were used? (recognized are 7 species of Pacific salmon plus Atlantic salmon)
Results – the abstract does not include any data
Conclusion, “These results definitely support…”; the use of “definitively” is not supported by the data from this laboratory study; and, the term does not seem consistent with the statement in line 206, “The trials were not extended any further because of a lack of interest in the subject by the salmon farms…”. Consider changing “These results definitely support the idea of using” to something like, “These results support the idea of exploring the value of”
Line 39 – I recommend replacing “pathologies” with “diseases”
Line 40 – replace “fryes” with “fry” (in English, “fry” is singular and plural)
Line 77, “In this work we propose to test the use of synthetic peptides…” – this seems like a line from a research proposal. Consider a revision to something like, “In this work we tested the use of synthetic peptides…”
Line 111 - MALDI-TOF mass spectrometry is mentioned here but not in the Methods section.
Figure 2 A and B – While this represents a nearly 100-fold difference between supernatant and monolayer, might this reflect a significant difference in infectivity? [Consider whether to address this idea in the Discussion] No error bars are shown: were replicates assayed to determine the variability of these estimates (not described in lines 290 – 292)? If no replicates were run, it decreased confidence in significance of the results.
Figure 3 caption – Please add the name of the virus used in the assay (IPNV).
Figure 3 – Line 336 describes 5 ponds with 250 fish, and Line 343 describes “56 days of evaluation” but the figure results from only the first 7 days, 4 groups, and states that n = 500. Can these apparent differences be clarified? Are the methods that support Figure 3 elsewhere?
Captions of Figure 4 and the first Figure 5 – Please add a description of what was tested for expression of VP2 gene. For first Figure 5, when were the controls tested?
First Figure 5 caption –this caption describes “two treatments”, but in the Methods Lines 365 and 370 refer to “the treatment”. Please clarify.
Second Figure 5 – needs to be renamed Figure 6
Lines 202 – 205, I recommend revising this paragraph to focus more on Discussion rather than repeating results. Also, I recommend replacing “proof” with “evidence” and “clearance” with “decrease in viral load”. Also, my understanding of the study is that “the peptide showed a significant decrease in VP2 expression” would be better stated as something like, “exposure to the peptide resulted in a significant decrease in VP2 expression”.
Lines 223 – 224, “This is another scenario in which the peptides could be used for clearance of the non-pathogenic variant of the virus” – given that the only two strains tested were HPR-delta, this statement is highly speculative. I recommend here changing to the wording to focus on how this study provides evidence that this might be effective, but actual testing with HPR0 strains is needed for confirmation.
Line 320, footnote 1 – change “Proffesor” to “Professor”
Line 336 – What type of pond was used? Lining (soil, concrete, etc.), volume, flow, temperature. What fish species was used?
Table 1. Please clarify the “stage” designation. At Stage 3, C2 x 3 h does not line up with any Pond (i.e., it is between Pond 4 and 5). Also, it is not clear how the four stages are reported in the Results section.
Line 344 – Please provide more information about how mortality was counted. Was mortality counted in all ponds and then later adjusted based on pond 5 mortality?
Line 347 – Thank you for identifying the fish species.
Line 349- How were the 84 fish analyzed for IPNV? PCR, tissue culture?
Are the “pre-smolt” in Line 347 the same as the “fry” in Line 352?
Line 350 – was GIM182 added to the water?
Line 353 – What type of ponds were used? Lining (soil, concrete, etc.), volume, flow, temperature.
Table 2 caption – consider changing the caption to “Scheme of weekly sampling (n) for fish treated and untreated with the GIM182 peptide.”
Line 361 – What fish species was used?
Line 363 – what size/shape tanks?
Line 368 – 371 – similar concept seems to be repeated in successive sentences (it seems that this is describing a treatment [line 370] rather than a bath [line 368])
Figure S2 caption – The first sentence implies that it is confocal microscopy that gets the peptides into the cell, which is not the case. Consider revising to something like:
“Figure S2. Confocal microscopy. Peptides in a cell. In red the peptide rhodamine; in green the peptide recognized by a specific antibody, the bright field is shown; and in yellow is the merge of the three layers.”
Figure S4. The caption here is very similar to the caption for the original Figure S2, but the images are different. Is this correct? Also, both images on the left column appear to be from the same microscopic field, but the image on the right is from a different microscopic field. Please clarify.
Author Response
Reviewer 2.
These experiments present interesting results from the use of peptides to decrease viral loads. If the efficacy of these peptides can be independently confirmed, they have the potential to provide value to the aquaculture industry. Indeed, in some areas of the world where ISAV is endemic, regulatory agencies do not automatically force ISAV-infected farms to depopulate (e.g., eastern Canada).
I offer several suggestions for revision to improve clarity of the presentation.
Abstract – Some of the recommended components of a Pathogens abstract are not clear here (see https://www.mdpi.com/journal/pathogens/instructions):
“Background: Place the question addressed in a broad context and highlight the purpose of the study” – Not clear to me in the abstract are the question addressed or the purpose of the study.
Methods: what species were used? (recognized are 7 species of Pacific salmon plus Atlantic salmon)
Results – the abstract does not include any data
Conclusion, “These results definitely support…”; the use of “definitively” is not supported by the data from this laboratory study; and, the term does not seem consistent with the statement in line 206, “The trials were not extended any further because of a lack of interest in the subject by the salmon farms…”. Consider changing “These results definitely support the idea of using” to something like, “These results support the idea of exploring the value of”
The reviewer is right, the abstract has been completely rewritten.
L17-29
Viral infections in salmonids represent an ongoing challenge for the aquaculture industry. Two RNA viruses, the infectious pancreatic necrosis (IPNV) virus and the infectious salmon anemia virus (ISAV), have become a latent risk without healing therapies available for either of them. In this context antiviral peptides emerge as effective and relatively safe therapeutic molecules.
Based on in silico analysis of VP2 protein from IPNV and the RNA-dependent RNA polymerase from ISAV, a set of peptides was designed and peptides were chemically synthesized to block selected key events in their corresponding infectivity processes. The peptides were tested in fish cell lines, in vitro, and four peptides were selected which were in decreasing order of viral load: peptide GIM182 for IPNV and peptides GIM535, GIM538 and GIM539 for ISAV. In vivo tests with the IPNV GIM 182 peptide were carried out using Salmo salar fish, showing significant decrease of viral load, and proving the safety of the peptide for fish.
The results indicate that the use of peptides as antiviral agents in disease control may be a viable alternative to explore in aquaculture.
Line 39 – I recommend replacing “pathologies” with “diseases”
R/ Done as recommended L44
Line 40 – replace “fryes” with “fry” (in English, “fry” is singular and plural)
R/ Done as indicated L45
Line 77, “In this work we propose to test the use of synthetic peptides…” – this seems like a line from a research proposal. Consider a revision to something like, “In this work we tested the use of synthetic peptides…”
R/ The paragraph was modified.
L78-88
With the aim to block viral replication, a key step in the life cycle of each virus was selected to be interfered with synthetic peptides: In IPNV, the homotrimer formed by VP2 protein as the initial step in capsid generation to interfere with the assembly process of newly-formed viruses; in ISAV the formation of the RNA-dependent-RNA polymerase (RdRp), requiring the assembly of three different subunits to interfere in the formation of a functional RdRP.
In this work synthetic peptides directed against the aforementioned key steps for each virus were tested, specifically designed and chemically synthesized. In order to accomplish these goals, in silico models were generated of the corresponding proteins and their complexes, to search for fruitful interactive regions able to generate peptide sequences to be synthesized and tested. As a result, one peptide from the VP2 protein of IPNV was tested in vitro and in vivo, and for ISAV three different peptides were selected from the RdRp complex and tested in vitro with positive results.
- Results
Line 111 - MALDI-TOF mass spectrometry is mentioned here but not in the Methods section.
R/ It is now included in Methods.
L355-361
Peptide identity was confirmed by MALDI-TOF mass spectrometry. 1mL of 1mg/mL peptide solution along with 1mL of matrix (10 mg/mL of a -cyano-4-hydroxycinnamic acid in 50% acetonitrile with 0.1% formic acid in water) was loaded onto a micro scout sample plate (Bruker Daltonics Inc., Billerica, MA, USA) and the measures were made in a MALDI-TOF-MS Microflex equipment (Bruker Daltonics Inc.) using flexControl 3.0 software (Bruker Daltonik GmbH, Bremen, Germany). Spectra were obtained in positive ion mode by reflection detection as additions of 10 rounds of 30 laser impacts each, at various points of the sample.
Figure 2 A and B – While this represents a nearly 100-fold difference between supernatant and monolayer, might this reflect a significant difference in infectivity? [Consider whether to address this idea in the Discussion] No error bars are shown: were replicates assayed to determine the variability of these estimates (not described in lines 290 – 292)? If no replicates were run, it decreased confidence in significance of the results.
R/ The measures were done in triplicate. The results obtained does not permit to conclude about infectivity; it will be necessary to infect new cells and titer the virus. Figure was changed, and the results are discussed with that focus.
L379 measurements were made in triplicate.
L248-257
In vitro assays showed that peptide GIM182 was able to decrease the number of copies of the virus in at least one log compared to the control, when the determination was made on the supernatant but not on the cell monolayer. The difference between the supernatant and the cell monolayer was, as expected, due to the technique used for virus detection (RT-PCR) that would include all the genetic material of the virus, and not necessarily complete virions, while in the supernatant only those particles that have been exported outside the cell and that constitute infectious viruses were detected. However, although a significant difference was observed between the controls and the application of the peptide, the degree of infectivity could only be determined by infecting new cells and determining the viral titer, as has been reported in similar works for the antiviral peptide test [23].
Figure 3 caption – Please add the name of the virus used in the assay (IPNV).
Figure 3 – Line 336 describes 5 ponds with 250 fish, and Line 343 describes “56 days of evaluation” but the figure results from only the first 7 days, 4 groups, and states that n = 500. Can these apparent differences be clarified? Are the methods that support Figure 3 elsewhere?
R/ Figure was changed; it is now Figure 4 and the assay is now better explained in M&M and in Results.
L143-156
The wet laboratory test was carried out in four different stages to unify the conditions of the treatments, and it was controlled for 56 days. In this time period, mortality was recorded on a daily basis, and was observed only during the first 24 days with the greatest difference between treatments observed in the first 10 days (Figure 4). The fish analyzed after the challenge were positive for IPNV, including those of the mortality control line.
The lowest mortality was recorded in the lines in which the peptide was added, both in the control peptide line, where there was no challenge with the virus, and in the pre-challenge and post-challenge test lines (T1: peptide added before the virus, and T2: peptide added after the virus). The statistical analysis of Mantel-Cox showed significant differences between all survival curves at a P value < 0.0005. A paired analysis showed that the greatest differences occurred between the control virus line, and the T1 along with the control peptide. T2 also showed a significant difference at a P value of 0.042 (supplementary Table S2). The three survival curves of the control peptide and the experimental lines of the treatments T1 and T2 lines did not show significant differences between them.
Captions of Figure 4 and the first Figure 5 – Please add a description of what was tested for expression of VP2 gene. For first Figure 5, when were the controls tested?
The captions of Figures 5 and 6 include what was tested. In the first case the RNA from fish kidney and gills, and in the second case the RNA extracted from the whole fish, as described in M&M section.
For Figure 5, now Figure 6, controls were taken at three times: at the start of the test, at 30 days and at 60 days. The values did not vary much over time for the infection control, and the value reported is the average of these controls with the standard deviation. This is clarified in the figure caption and in the text.
First Figure 5 caption –this caption describes “two treatments”, but in the Methods Lines 365 and 370 refer to “the treatment”. Please clarify.
Caption was corrected, it was one treatment (peptide bath at 6, 9 and 12 days) and analyzed at two different times, 30 and 60 days.
Second Figure 5 – needs to be renamed Figure 6
R/ Done as indicated
Lines 202 – 205, I recommend revising this paragraph to focus more on Discussion rather than repeating results. Also, I recommend replacing “proof” with “evidence” and “clearance” with “decrease in viral load”. Also, my understanding of the study is that “the peptide showed a significant decrease in VP2 expression” would be better stated as something like, “exposure to the peptide resulted in a significant decrease in VP2 expression”.
R/ Thanks for the suggestion, the paragraph was rewritten.
L274-282
In the third in vivo assay, the application of the peptide was carried out by bathing the fish three times, on days 6, 9 and 12, and samples were taken for analysis at 30 and 60 days. The results of this trial showed significant differences in the expression of IPNV VP2, when comparing the fish that were exposed to the peptide, versus those in the control tanks. In addition, a greater decrease in the viral load was observed in the samples analyzed at 60 days, which suggests a cumulative effect of the peptide since, unlike than in the previous tests, a repeated application was carried out three times.
This is evidence that in persistent infections the exposure to the peptide decreases the viral load, so reducing the risk of infection dissemination by these fish.
Lines 223 – 224, “This is another scenario in which the peptides could be used for clearance of the non-pathogenic variant of the virus” – given that the only two strains tested were HPR-delta, this statement is highly speculative. I recommend here changing to the wording to focus on how this study provides evidence that this might be effective, but actual testing with HPR0 strains is needed for confirmation.
R/ The paragraph was complemented, and although the reviewer is right in the sense that it is necessary to test with HPR0 to confirm the effect, the high conservation of the RdRp between strains suggests that the effect may be transversal regardless of the strain, but this needs to be experimentally tested. The current limitation is that this strain is not cultivable in the laboratory and therefore it would be necessary to perform the tests in vivo.
L302-308
The results obtained, shows that the peptides were able to decrease the viral load with the two variants tested, taking into account that they are directed against the RdRdp, and that this is highly conserved protein among the different virus variants, including the non-pathogenic HPR0. There is then a high probability that the peptides can be applied against HPR0 variant as well. However, up to now it has not been possible to cultivate the HPR0 strain in the laboratory, so it seems appropriate to carry out in vivo tests with fish infected with this variant to confirm it.
Line 320, footnote 1 – change “Proffesor” to “Professor”
R/ The footnote was removed; it is corrected and included in the acknowledgments
Line 336 – What type of pond was used? Lining (soil, concrete, etc.), volume, flow, temperature. What fish species was used?
R/ Our apologies for the omissions; information is now included in all applicable assays.
Table 1. Please clarify the “stage” designation. At Stage 3, C2 x 3 h does not line up with any Pond (i.e., it is between Pond 4 and 5). Also, it is not clear how the four stages are reported in the Results section.
R/ Some of the stages are common for some tanks, for example the C2x3h is common for peptide control and mortality control; however, to clarify this we repeated the condition for each tank and made clear which are the common stages in the text.
L143-144
The wet laboratory test was carried out in four different stages to unify the conditions of the treatments.
L429-430
Two treatments in four different stages were tested, with stages 2 and 4 being common for all tanks, and three types of controls were used according to the distribution shown in Table 2.
Line 344 – Please provide more information about how mortality was counted. Was mortality counted in all ponds and then later adjusted based on pond 5 mortality?
R/ Mortality was recorded on a daily basis, the methodology is better explained and we used now survival curves with all the data without adjustment.
L435-436
The total trial lasted 67 days, 11 days of acclimatization and 56 days of evaluation. Mortality was recorded on a daily basis for all the tanks used in the assay.
Line 347 – Thank you for identifying the fish species.
R/ Our apologies for the omissions; information is now included in all applicable assays.
Line 349- How were the 84 fish analyzed for IPNV? PCR, tissue culture?
R/ Fish were analyzed by PCR, from RNA extracted from kidney and gills.
L441-442
Prior to transfer, 84 of these fish were analyzed to determine the presence of IPNV by RT-PCR in RNA extracted from fish kidney and gills.
Are the “pre-smolt” in Line 347 the same as the “fry” in Line 352?
R/ Yes they are; the terms were changed.
Line 350 – was GIM182 added to the water?
Yes; now this is included in the text
L443-444
During the transfer, peptide GIM182 was added to the water to one of the groups at a concentration of 10-7 M and the other group was kept as control.
Line 353 – What type of ponds were used? Lining (soil, concrete, etc.), volume, flow, temperature.
R/ Our apologies for the omissions; information is now included in all applicable assays.
Table 2 caption – consider changing the caption to “Scheme of weekly sampling (n) for fish treated and untreated with the GIM182 peptide.”
R/ The caption of the table was changed to: Number of fish sampled weekly in treated and control tanks.
Line 361 – What fish species was used?
Line 363 – what size/shape tanks?
R/ Our apologies for the omissions; information is now included in all applicable assays.
Line 368 – 371 – similar concept seems to be repeated in successive sentences (it seems that this is describing a treatment [line 370] rather than a bath [line 368])
R/ We use the term treatment to refer to the application of the peptide, which was done by bath.
Figure S2 caption – The first sentence implies that it is confocal microscopy that gets the peptides into the cell, which is not the case. Consider revising to something like:
“Figure S2. Confocal microscopy. Peptides in a cell. In red the peptide rhodamine; in green the peptide recognized by a specific antibody, the bright field is shown; and in yellow is the merge of the three layers.”
Figure S4. The caption here is very similar to the caption for the original Figure S2, but the images are different. Is this correct? Also, both images on the left column appear to be from the same microscopic field, but the image on the right is from a different microscopic field. Please clarify.
R/ Figures are now included in the text and captions were corrected.
Reviewer 3 Report
The currently manuscript entitled ‘Interfering synthetic peptides as a novel tool to control viral infections in fish’ describes the use of peptides based on virus sequence to block virus production theoretically by blocking protein assembly. This is a fascinating new line of research; however, the manuscript requires major modifications prior to publication.
Major
- A control peptide of similar length but of unrelated sequence is needed, even if just in the in vitro trials. This would ensure that the results demonstrated are due to those specific peptides and not just peptide introduction in general. This is particularly important to support the proposed mechanism of action (Line 211-212).
- I would like to see the cell association figures in the body of the manuscript. The cell association patterns are different between the peptides, and this should be discussed. Specifically, the IPNV peptide appears to be perinuclear, perhaps in the ER?
- Also, it is important to report the amount of peptide/cell, the association efficiency. It may be different, as there is only 1 cell in Figure S2, thus how many cells in a larger field of view took up the peptide?
- Also, the authors need to use the words 'cell association' instead of 'cell entry' as they cannot rule out with the assays used that the peptides are on the cell surface instead of inside the cell
- I would also like to see in the discussion a section on proposed peptide entry mechanisms
- Re-draft the results section. The purpose of the results section is to report the results of the study in words as well as in figures. This includes:
- Describing briefly how the animal studies were set up and how the differ from each other
- Describe the results demonstrated in the figures in words, to help the reader interpret the figures, instead of simply referring to the figure.
- Missing data. There are a number of places where results are reported, but no data is shown
- 1.3 Line 110-112. Please include data or cite data not shown
- Line 113-115: efficacy of specific peptides, this data should be included, if only in the supplementary data
- The peptides are claimed to be non-toxic but no data is shown. Either include the data or remove the cytotoxicity assay from the methods section.
Minor
- The title reads like a review paper, please make it more specific to the present study
- Remove ‘definitely’ from line 25
- Line 47: Include kDa size of VP5
- Remove ‘the’ from line 56: ISAV is the agent of the Infectious salmon anemia virus
- Line 58: change economics to economic
- Remove ‘an’ from line 59: has improved thanks to an adequate
- Remove ‘the’ in line 64: Structurally, the virus is similar to the influenza virus
- Line 73-75: it is important to note that these are two steps in virus replication that require protein assembly but they are not the only ones or are they necessarily the most important, they are the steps that the authors chose to focus upon.
- Add an ‘A’ in line 91: A homology model was obtained
- Figure 1: the ball and stick regions are not readily visible, could arrows be used to make them more identifiable to the reader?
- Figure S2: phase picture is too dark.
- Line 111: remove ‘of them’
- Figure 2 legend is not readable, please fix.
- Figure 3: report peptide concentration
- Figure 5. Can you please identify where the peptides used in this study can be found in the structural model?
- Methods 4.3 does not include the antibody used in Fig S2 in the methods
Author Response
First of all we want to thank the reviewers for their extensive review and enriching comments.
We also want to apologize for the mistakes and oversights, and we hope that the current version of the manuscript complies with the requested revisions.
Reviewer 3
The currently manuscript entitled ‘Interfering synthetic peptides as a novel tool to control viral infections in fish’ describes the use of peptides based on virus sequence to block virus production theoretically by blocking protein assembly. This is a fascinating new line of research; however, the manuscript requires major modifications prior to publication.
Major
- A control peptide of similar length but of unrelated sequence is needed, even if just in the in vitro trials. This would ensure that the results demonstrated are due to those specific peptides and not just peptide introduction in general. This is particularly important to support the proposed mechanism of action (Line 211-212).
R/ Preliminary test was made including two unrelated peptides, GIM182 from IPNV and an analogue of this peptide GIM1094. This is now included in the manuscript, and the respective figure is included in supplementary material.
L216-223
A preliminary test was performed with the peptides designed for the ISAV RdRdP, and additionally two unrelated peptides were included: the IPNV GIM182 peptide and an analogue of it, the peptide GIM1094. The tests were performed at two concentrations, 10-3 and 10-4 M, and two different times, 12 and 24 hours, using the HPR14a strain (Supplementary Figure S6). Although the results were not significant (only a significant difference was observed between the GIM1094 peptide and the others at a concentration of 10-3 M at 24 hours), this assay allowed to determine that the effect of the peptides is specific and that the used concentrations produced a decrease in the viral load. The following tests were performed only with the RdRP peptides.
L287-288
Preliminary tests showed the specificity of the chosen peptides in decreasing the viral load in ISAV (supplementary Figure S5).
- I would like to see the cell association figures in the body of the manuscript. The cell association patterns are different between the peptides, and this should be discussed. Specifically, the IPNV peptide appears to be perinuclear, perhaps in the ER? 1. Also, it is important to report the amount of peptide/cell, the association efficiency. It may be different, as there is only 1 cell in Figure S2, thus how many cells in a larger field of view took up the peptide?
R/ The figures are now in the body of the manuscript (figures 2 and 8) and other supplementary figures are included where a larger field can be observed; also a confocal Z-stack video was included that shows the entry of the peptide GIM182 into the cells.
- Also, the authors need to use the words 'cell association' instead of 'cell entry' as they cannot rule out with the assays used that the peptides are on the cell surface instead of inside the cell
R/ We changed the term as suggested by the reviewer, however as can be seen in the confocal microscopy Z-stack video of GIM182 peptide, the peptide is observed inside the cells and not in the surface.
- I would also like to see in the discussion a section on proposed peptide entry mechanisms
With the tests carried out, it is not possible to propose a mechanism for the entry of the peptides into the cell, especially bearing in mind that they are peptides with different sequences, and not associated with a specific pattern (in charge or hydrophobicity, for example).
- Re-draft the results section. The purpose of the results section is to report the results of the study in words as well as in figures. This includes: 1. Describing briefly how the animal studies were set up and how the differ from each other
- Describe the results demonstrated in the figures in words, to help the reader interpret the figures, instead of simply referring to the figure.
R/ The section of Results has been rewritten in several parts as can be seen highlighted in the manuscript.
- Missing data. There are a number of places where results are reported, but no data is shown 2. 1.3 Line 110-112. Please include data or cite data not shown
R/ Data not shown is now specified in the text.
- Line 113-115: efficacy of specific peptides, this data should be included, if only in the supplementary data
R/ The preliminary test are now included in the text and as Supplementary Figure S2
L126-129
The initial evaluation was performed with de novo infection of the CHSE-214 cell line with IPNV with four synthetic peptides and only one of them, GIM182 (TSDLPTSKAWG), significantly decreased viral replication compared with the infection control (Supplementary Figure S3); accordingly, the subsequent assays were performed with this peptide.
- The peptides are claimed to be non-toxic but no data is shown. Either include the data or remove the cytotoxicity assay from the methods section.
R/ The cytotoxicity assay was removed from the Materials and Methods section.
Minor
- The title reads like a review paper, please make it more specific to the present study
R/ Changed to : Synthetic peptides as a promising alternative to control viral infections in salmonid fish
Remove ‘definitely’ from line 25
R/Abstract was completely rewritten
- Line 47: Include kDa size of VP5
R/ Done as indicated
- Remove ‘the’ from line 56: ISAV is the agent of the Infectious salmon anemia virus
R/ Done as indicated
- Line 58: change economics to economic
R/ Done as indicated
- Remove ‘an’ from line 59: has improved thanks to an adequate
R/ Done as indicated
- Remove ‘the’ in line 64: Structurally, the virus is similar to the influenza virus
R/ Fragment has been changed
L68-72
ISAV is a single stranded RNA virus of negative polarity, with eight segments encoding at least ten proteins [8,9]: two glycoproteins (F and HE) encoded by segments 5 and 6 form the viral capsid, the proteins PB2, PB1 and PA encoded by segments 1, 2 and 4 respectively, which are the subunits of the RNA dependent RNA polymerase (RdRP) that, together with the nucleoprotein (NP) encoded by segment 3, form the ribonucleic protein complex.
- Line 73-75: it is important to note that these are two steps in virus replication that require protein assembly but they are not the only ones or are they necessarily the most important, they are the steps that the authors chose to focus upon.
R/ The reviewer is right, the paragraph was rewritten
L78-82
With the aim to block viral replication, a key step in the life cycle of each virus was selected to be interfered with synthetic peptides: In IPNV, the homotrimer formed by VP2 protein as the initial step in capsid generation to interfere with the assembly process of newly-formed viruses; in ISAV the formation of the RNA-dependent-RNA polymerase (RdRp), requiring the assembly of three different subunits to interfere in the formation of a functional RdRP.
- Add an ‘A’ in line 91: A homology model was obtained
R/ Done as indicated
- Figure 1: the ball and stick regions are not readily visible, could arrows be used to make them more identifiable to the reader?
R/ Figure has been improved following the suggestion made
- Figure S2: phase picture is too dark.
R/ Figure was improved
- Line 111: remove ‘of them’
R/ Done as indicated
Figure 2 legend is not readable, please fix.
R/ Done as indicated
- Figure 3: report peptide concentration
R/ Done as indicated
Figure 5. Can you please identify where the peptides used in this study can be found in the structural model?
R/ Done as indicated
- Methods 4.3 does not include the antibody used in Fig S2 in the methods
R/ It is now included.
L368-370
Additionally, the production of a polyclonal antibody was requested from the company Pickcell®, The Netherlands, obtained in rabbit and that was used to check the cell association of the peptide GIM182 without rhodamine into the CHSE-214 cells.
Round 2
Reviewer 1 Report
Manuscript: Synthetic peptides as a promising alternative to control viral infections in salmonid fish, by Constanza Cárdenas et al.
This is a second version of a manuscript already evaluated by myself and, I must recognize that it has been really improved. Unfortunately, it still needs important corrections.
Throughout the test, in the attached pdf, I have written some corrections to improve the language and/or the comprehension of some paragraphs and sentences. Please, Take a look and consider if they my corrections are correct.
In addition, I have included many comments and request for the authors to consider. Please, read them carefully and apply or justify why it should not be applied.
I still believe that the study deserves to be published. However, it cannot be accepted in its present form. Therefore, I encourage the authors to improve it, perhaps following my suggestions.
What do you want to do ? New mailCopy What do you want to do ? New mailCopy What do you want to do ? New mail
Author Response
We want to thank again for the comments of the reviewers, we have answered all the comments and we hope that the present version of the manuscript complies with the requested revisions.
Below is a detailed list of all the changes made, according to each reviewer
Reviewer 1.
Thanks for all the corrections made in the pdf file, they were all made, and the comments are answered below:
L456 The in vitro assays showed that peptide GIM182 was able to reduce the number of viral copies in at least one log compared to the control, when the measurement was performed on the supernatant but not on the cells.
Comment: Could this suggest that the peptides affect morphogenesis and/or release? It seems probable: RNA production (intracellular) seems nos to be affected, whereas extracellular RNA is.
R/ It is difficult to affirm this, because of the bias in the detection technique. We think that the peptides indeed affects virus morphogenesis, and in a previous thesis work it was possible to see some defective viral particles, but this need to be confirmed.
L486 In addition, a greater decrease in the viral load was observed in the samples analyzed at 60 days, which suggests a cumulative effect of the peptide since, unlike than in the previous tests, a repeated application was carried out three times.
Comment: I don´t understand this sentence; please improve it.
R/ It was rewritten as follows:
In addition, a greater decrease in the viral load was observed in the samples analyzed at 60 days, which suggests a cumulative effect due to the repeated application of the peptide.
L500 The results suggest that the interfered process is the replication, as can be seen in the reduction of the expression of the genomic RNA (Figure 9).
Comment: Figure 6 .This Fig does not show what the authors state. In fact, I did not find those results.
R/ Sorry, is Figure 9, we think that the interfered process is replication, because the primers used for detection are specific for genomic RNA, and the amplification with the primers for the transcription products did not yield positive results, we include a phrase to highlight this:
L325
These results were observed with the specific primers directed against the genomic RNA, but not against the transcription products (results not shown)
Comment: Material and methods: viruses and cell lines should be described
R/ Done as recommended. Point 4.3 in material and methods.
L511 The results obtained in our study, show that the peptides are able to decrease the viral load of the variants tested (HPR3a and HPR14a). Considering that they are directed against the RdRdp, which is a highly conserved protein among the different virus variants (including the non-pathogenic HPR0), there is then a high probability that the peptides could be also applied against HPR0 variant.
Comment: The strains used should have been described, and they are not. Therefore, I cannot be sure if they are actually representative of both variants.
R/ We are not sure to what the reviewer refers to with this comment, the two variants used correspond to deleted variants of the virus, the non-pathogenic or HPR0 variant is not cultivable in the laboratory, therefore it cannot be used for the assays.
L595 IPNV expression was measured by VP1 and VP2 mRNAs amplification using RT-qPCR real time procedure, and ELF-1a as housekeeping gene (Stratagene 1 step RT-qPCR Kit) with specific primers for each segment (Table S4)
Comment: Are these primers really specific for mRNA? This should be clarified
R/ The reviewer is correct, the primers are not specifically for the mRNA, it was corrected in the manuscript.
L629 In vivo assays
Comment: Are there regulations to ensure the good animal practices and experimental animal welfare in Chile? In Europe, it is mandatory indicateion of the regulations followed and permissions obtained to experiment with fish. Should it be indicated also here?
R/ We include the point 4.8 with the Ethics Statement
Comment: Conclusions
This is interesting and well written, but it is more appropriate for Introduction or Discussion. Conclusion should be, to my understanding, more concise.
R/ Although we respect the opinion of the reviewer, we think that the conclusions in their current form provide our point of view on the subject.
Reviewer 2 Report
This manuscript is significantly improved from the first version. However, several points still require clarification. I offer comments and suggestions to improve the manuscript:
Title – My understanding is that all tests were done with Atlantic salmon; therefore, I recommend replacing “salmonid fish” with “Atlantic salmon”
Abstract – If the species of fish tested is not listed in the title, it needs to be listed here.
Line 36, “the second export product” – more detail is needed here; “the second most valuable export product…”?
Lines 77 – 81 – This is a key concept of the paper. I find it difficult to get the main point(s) out of this 74-word sentence. I recommend breaking the paragraph into at least three separate sentences. Also, the current wording describes “the homotrimer formed by VP2 protein” as a step, but it seems more like a thing than a step. Can this be clarified?
Line 83 – I recommend listing these steps in chronological order. [Am I correct that this would be design, synthesis, and then testing? Also, the first sentence of this paragraph describes what was done (methods), but the second sentence refers to what was done as goals; perhaps “goals” should be replaced with something like “tasks”.] Alternatively, the wording could be changed to present the goal(s) of the study.
Line 113 mentions “supplementary video GIM-182”, but this is not included in supplementary files available for my review.
Line 126 and Figure S3 – the text says that only GIM182 significantly decreased viral replication, but the figure also shows a significant change for GIM176 at both concentrations. Please clarify. Also, in the Figure S3 caption, does ** designate P>0.001 (as written) or P<0.001 (which is more consistent with standard reporting for P)?
Figure 3 – What do the error bars represent (SE, STDEV, etc.)?
Lines 142 – 146 – what was done here is not clear. What was done from day 24 – 56? Did mortality occur during this time that was not observed? Lines 430 – 431 include, “The total trial lasted 67 days, 11 days of acclimatization and 56 days of evaluation. Mortality was recorded on a daily basis for all the tanks used in the assay.”
Line 144 – Can “was observed” be replaced with “occurred”?
Figure 4. This figure is very different from the original Figure 3. Are the axes correct on this figure (particularly B)? Figure 3A seems to show that mortality of the controls was greater than any of the experimental groups. If correct, then the assay is invalid because non-protocol mortality was greater than experimental morality. If I am not correct, please clarify. For example, were all fish used in this experiment persistently infected with IPNV? Also, the caption does not explain figure B.
Line 165 – Replace “The results showed a decrease in the expression…” with “The results showed no significant change in the expression…”
Line 173 – It is not clear in the Methods if all fish were persistently infected with IPNV. If not, then replace “In the third in vivo assay with persistently infected fish…” with “In the third in vivo assay, which used persistently infected fish…”
Line 176 – replace “realtive” with “relative”
Line 178, “no mortality was observed.” – How much mortality occurred that was not observed? If none, consider changing this wording to “no mortality occurred.”
Figure 5 (line 170) – according to Table 3 (line 448), fish were sampled 4 different times. Which results (all?) are reported here? According to Table 3, the tank 7 is a control, but Figure 5 has tank 7 as a peptide. According to Table 3, tank 5 is a peptide, but Figure 5 has tank 5 as a control. Please clarify. What do the error bars represent (SE, STDEV, etc.)?
Figure 6 – still not clear is which controls are reported here (line 463 says that controls were sampled at 0, 30, and 60 days). Consider reporting controls from each sample date separately. Also, I find confusing the designation of T1 and T2 in Figures 4 and 6 seem to refer to different things. I recommend changing the designation in Figure 4 from T to TL or L. What do the error bars represent (SE, STDEV, etc.)?
Line 218, “results were not significant” vs. Line 220 “the used concentrations produced a decrease in the viral load” – Please clarify the relation between non-significant results and the reported decrease in the viral load. Are these different endpoints?
Figure 9 – What concentration of peptide was used? What do the error bars represent (SE, STDEV, etc.)?
Lines 268 -269 – Replace “showed only a slight difference” with “showed no significant difference” [this information can then be deleted from the following sentence.]
Lines 272 – 277 – this paragraph contains 108 words, 100 of which are repetition of results and only 8 are Discussion (“which suggests a cumulative effect of the peptide”). I recommend revision to focus more on the Discussion that on repetition of the results.
Line 285 – Should this refer to Figure S6? This sentence is a repetition of Results (consider deleting).
Line 290, “genomic RNA” – other double stranded RNA viruses occur in the cell in two forms, mRNA (single-stranded) and gRNA (double-stranded). Is IPNV such a virus? If so, has the analysis detected both forms or just gRNA? Please revise accordingly.
Lines 293 – 310 – I like this Discussion
Lines 433 - 437– This might need to be clarified… Were 150,000 fish placed in one 3 m3 tank and another 150,000 fish placed in another 3 m3 tank? Were the 84 fish sampled as 42 fish from one tank and 42 fish from the other tank? Were fish randomly distributed into the transport tanks, or had they been reared in separate tanks at the source farming center?
Lines 458 - 462 – Not clarified from the original version, but the two sentences here seem to say nearly the same thing. Can they be combined into a single sentence? Can “finished” be deleted from line 461?
Line 484 – replace “mainly due to the fact that” with “mainly because”
Line 498 – replace “powerful tools of fresh air” with “powerful tools”
Author Response
We want to thank again for the comments of the reviewers, we have answered all the comments and we hope that the present version of the manuscript complies with the requested revisions.
Below is a detailed list of all the changes made, according to each reviewer
Reviewer 2
Title – My understanding is that all tests were done with Atlantic salmon; therefore, I recommend replacing “salmonid fish” with “Atlantic salmon”
Abstract – If the species of fish tested is not listed in the title, it needs to be listed here.
R/ We changed the title as suggested by the reviewer, but also in the abstract, L26 appears Salmo salar
Line 36, “the second export product” – more detail is needed here; “the second most valuable export product…”?
R/ Done as indicated.
Lines 77 – 81 – This is a key concept of the paper. I find it difficult to get the main point(s) out of this 74-word sentence. I recommend breaking the paragraph into at least three separate sentences. Also, the current wording describes “the homotrimer formed by VP2 protein” as a step, but it seems more like a thing than a step. Can this be clarified?
R/ The paragraph was modified.
With the aim to block viral replication, a key step in the life cycle of each virus was selected to be interfered with synthetic peptides. In IPNV, the homotrimer formed by the VP2 protein, which seems to be the building block in capsid generation, to interfere with the assembly process of newly-formed viruses. In ISAV the formation of the RNA-dependent-RNA polymerase (RdRp), requiring the assembly of three different subunits, to interfere in the formation of a functional RdRP.
Line 83 – I recommend listing these steps in chronological order. [Am I correct that this would be design, synthesis, and then testing? Also, the first sentence of this paragraph describes what was done (methods), but the second sentence refers to what was done as goals; perhaps “goals” should be replaced with something like “tasks”.] Alternatively, the wording could be changed to present the goal(s) of the study.
R/ Rewritten as indicated:
In this work peptides specifically designed against the aforementioned key steps for each virus were chemically synthesized and tested. In order to accomplish the interference of each viral cycle, in silico models of the corresponding proteins and their complexes were generated to search for fruitful interactive regions able to generate peptide sequences to be synthesized and tested.
Line 113 mentions “supplementary video GIM-182”, but this is not included in supplementary files available for my review.
R/ Sorry for the inconvenience, there were problems uploading the video to the system, so it was sent to the editor by mail who handled this part.
Line 126 and Figure S3 – the text says that only GIM182 significantly decreased viral replication, but the figure also shows a significant change for GIM176 at both concentrations. Please clarify. Also, in the Figure S3 caption, does ** designate P>0.001 (as written) or P<0.001 (which is more consistent with standard reporting for P)?
R/ Thanks for the indications, the legend of the figure was corrected and we changed the paragraph to clarify:
The initial evaluation was performed with de novo infection of the CHSE-214 cell line with IPNV with four synthetic peptides and two of them, GIM176 (YRWNLNQTALEFD) and GIM182 (TSDLPTSKAWG), significantly decreased viral replication compared with the infection control (Supplementary Figure S3); however peptide GIM182 showed higher conservation between IPNV genogroups (Supplementary Figure S1) and, accordingly, the subsequent assays were performed with this peptide.
Figure 3 – What do the error bars represent (SE, STDEV, etc.)?
R/ Our apologies for the omission. All the error bars in the different figures represent standard deviations, they were already included in the respective legends.
Lines 142 – 146 – what was done here is not clear. What was done from day 24 – 56? Did mortality occur during this time that was not observed? Lines 430 – 431 include, “The total trial lasted 67 days, 11 days of acclimatization and 56 days of evaluation. Mortality was recorded on a daily basis for all the tanks used in the assay.”
R/ In this assay from day 24-56 no mortality occurred. This assay yielded unexpected results, and as the reviewer mentions the assay is not valid from a statistical point of view. We think that the behavior was due to an initial outbreak of IPNV, therefore mortality occurred in all cases. However, as it is in the manuscript, the assay was valuable from other approaches.
The experimental challenge in the wet lab yielded an unexpected result; first, the mortality control line behaved like the virus control and although all the tanks were kept in the same conditions, the fish included in the trial had a pre-existing infection with IPNV, which could have caused these mortalities; second, mortality occurred only during the first 24 days (Figure 4). Despite this, it was possible to observe significant differences between the survival curves of virus control and the fish that were exposed to the peptide.
Line 144 – Can “was observed” be replaced with “occurred”?
R/ Done as recommended.
Figure 4. This figure is very different from the original Figure 3. Are the axes correct on this figure (particularly B)? Figure 3A seems to show that mortality of the controls was greater than any of the experimental groups. If correct, then the assay is invalid because non-protocol mortality was greater than experimental morality. If I am not correct, please clarify. For example, were all fish used in this experiment persistently infected with IPNV? Also, the caption does not explain figure B.
R/ The figure was replaced with the correct axes, and the caption was also corrected. Regarding the assay, please see the above answer on comment about Lines 142-146
Line 165 – Replace “The results showed a decrease in the expression…” with “The results showed no significant change in the expression…”
R/ Done as recommended.
Line 173 – It is not clear in the Methods if all fish were persistently infected with IPNV. If not, then replace “In the third in vivo assay with persistently infected fish…” with “In the third in vivo assay, which used persistently infected fish…”
R/ Rewritten as indicated:
In the third in vivo assay, also carried out with persistently infected fish, the relative expression of IPNV VP2 was compared between controls and samples from a pool of fry fish at 30 and 60 days after the addition of the peptide to the water.
Line 176 – replace “realtive” with “relative”
R/ Done as recommended.
Line 178, “no mortality was observed.” – How much mortality occurred that was not observed? If none, consider changing this wording to “no mortality occurred.”
R/ Done as recommended.
Figure 5 (line 170) – according to Table 3 (line 448), fish were sampled 4 different times. Which results (all?) are reported here? According to Table 3, the tank 7 is a control, but Figure 5 has tank 7 as a peptide. According to Table 3, tank 5 is a peptide, but Figure 5 has tank 5 as a control. Please clarify. What do the error bars represent (SE, STDEV, etc.)?
R/ Thanks for the corrections. The tanks were wrongly marked on the table, they have been corrected. The results presented correspond to the average of the last two samplings. The previous samplings are not presented because they did not present any difference. This was clarified in the manuscript.
Figure 5 shows the results corresponding to the average of the last two samplings, the data corresponding to the first two weeks did not show any difference. The results showed no significant change in the expression of IPNV VP2 in the fish that were exposed to the peptide compared to those that were in the control tanks; although the difference was not significant, a tendency to decrease the viral load was observed (Figure 5).
Figure 6 – still not clear is which controls are reported here (line 463 says that controls were sampled at 0, 30, and 60 days). Consider reporting controls from each sample date separately. Also, I find confusing the designation of T1 and T2 in Figures 4 and 6 seem to refer to different things. I recommend changing the designation in Figure 4 from T to TL or L. What do the error bars represent (SE, STDEV, etc.)?
R/ Thanks for the suggestions, we changed figure 6, now including all controls, and we also changed the legend of figure 4, as suggested by the reviewer.
Line 218, “results were not significant” vs. Line 220 “the used concentrations produced a decrease in the viral load” – Please clarify the relation between non-significant results and the reported decrease in the viral load. Are these different endpoints?
R/ Significant is referred to statistical analysis, and the decrease reported is the tendency of the results. The paragraph was changed to clarify this:
Although the results were, in general, not significant (only a significant difference was observed between the GIM1094 peptide and the others at a concentration of 10-3 M at 24 hours), the results in this assay suggested that the effect of the peptides is specific and that the concentrations used produced a decrease in the viral load.
Figure 9 – What concentration of peptide was used? What do the error bars represent (SE, STDEV, etc.)?
R/ Our apologies for the omission. Figure legend was corrected including peptide concentration, and error bars meaning.
Lines 268 -269 – Replace “showed only a slight difference” with “showed no significant difference” [this information can then be deleted from the following sentence.]
R/ Done as recommended.
Lines 272 – 277 – this paragraph contains 108 words, 100 of which are repetition of results and only 8 are Discussion (“which suggests a cumulative effect of the peptide”). I recommend revision to focus more on the Discussion that on repetition of the results.
R/ Rewritten as indicated:
The results for the third in vivo assay showed a decrease in the viral load for the fish exposed to the peptide, with the greatest effect on samples analyzed at 60 days, which suggests a cumulative effect due to the repeated application of the peptide.
Line 285 – Should this refer to Figure S6? This sentence is a repetition of Results (consider deleting).
R/ Done as recommended.
Line 290, “genomic RNA” – other double stranded RNA viruses occur in the cell in two forms, mRNA (single-stranded) and gRNA (double-stranded). Is IPNV such a virus? If so, has the analysis detected both forms or just gRNA? Please revise accordingly.
R/ We are not sure to what the reviewers refers to with this comment. This paragraph refers to ISAV, whose ssRNA is of negative polarity, making a difference between replication and transcription. In the assays performed, this difference permit us to detect genomic RNA thanks to specific primers.
Lines 293 – 310 – I like this Discussion
R/Thanks
Lines 433 - 437– This might need to be clarified… Were 150,000 fish placed in one 3 m3 tank and another 150,000 fish placed in another 3 m3 tank? Were the 84 fish sampled as 42 fish from one tank and 42 fish from the other tank? Were fish randomly distributed into the transport tanks, or had they been reared in separate tanks at the source farming center?
R/ The two groups of 150000 fish were randomly distributed in the transportation tanks at a density of 40 kg/m3, using 50 tanks for the whole group (300000 fish). The 84 his were sampled randomly from the initial group. The paragraph were modified to calrify this:
An initial group of 300,000 20 g Salmo salar pre-smolt were transferred between two farming centers. The transport was carried out in two groups (150,000 each) at a density of 40 kg/m3 and a temperature of 8-12 ° C, in cubic fiberglass reinforced plastic transport tanks of 3 m3 (50 tanks in total), with oxygen supplementation. Prior to transfer, 84 of these fish, randomly sampled from the initial group, were analyzed to determine the presence of IPNV by RT-qPCR applied to RNA extracted from fish kidney and gills.
Lines 458 - 462 – Not clarified from the original version, but the two sentences here seem to say nearly the same thing. Can they be combined into a single sentence? Can “finished” be deleted from line 461?
R/ Done as recommended.
Line 484 – replace “mainly due to the fact that” with “mainly because”
Line 498 – replace “powerful tools of fresh air” with “powerful tools”
R/ Done as recommended.
Reviewer 3 Report
All the changes are appropriate. Just one comment still. Figure S6 GIM182 and GIM1094 are the control peptides, but it appears at 10-3M at 12h they had an effect. Could the authors please cite this in the manuscript and discuss how non-specific peptides could be affecting virus replication.
Author Response
We want to thank again for the comments of the reviewers, we have answered all the comments and we hope that the present version of the manuscript complies with the requested revisions.
Below is a detailed list of all the changes made, according to each reviewer
Reviewer 3
All the changes are appropriate. Just one comment still. Figure S6 GIM182 and GIM1094 are the control peptides, but it appears at 10-3M at 12h they had an effect. Could the authors please cite this in the manuscript and discuss how non-specific peptides could be affecting virus replication.
R/ Thanks for the comments. Regarding figure S6, with only this preliminary test, and also taking into account the high dispersion of the data, we dare not to discuss about the effect that these peptides may have on the virus.
Round 3
Reviewer 1 Report
Let me give my opinion on the format of the Conclusions section, which the authors have justified why they do not modify as I indicated: With respect to that chapter, I must reaffirm myself in saying that it is not a real conclusion of a research work; could it be acceptable as Conclusion in a review? Sure it would. However, in a normal research paper, the conclusion must be concise and direct. But, of course, it is my personal opinion.